# Targeting netrin-3 in small cell lung cancer and neuroblastoma

Shan Jiang[1,†], Mathieu Richaud[1,†], Pauline Vieugué[1,†], Nicolas Rama[1] (iD), Jean-Guy Delcros[1,2] (iD), Maha Siouda[3], Mitsuaki Sanada[4], Anna-Rita Redavid[1], Benjamin Ducarouge[5], Maëva Hervieu[1], Silvia Breusa[1], Ambroise Manceau[1], Charles-Henry Gattolliat[6], Nicolas Gadot[7], Valérie Combaret[7], David Neves[5], Sandra Ortiz-Cuaran[3], Pierre Saintigny[3], Olivier Meurette[1] (iD), Thomas Walter[1,8], Isabelle Janoueix-Lerosey[9], Paul Hofman[10], Peter Mulligan[3] (iD), David Goldshneider[5], Patrick Mehlen[1,3,*,‡] (iD) & Benjamin Gibert[1,3,*,‡] (iD)

## Abstract

The navigation cue netrin-1 is well-documented for its key role in cancer development and represents a promising therapeutic target currently under clinical investigation. Phase 1 and 2 clinical trials are ongoing with NP137, a humanized monoclonal antibody against netrin-1. Interestingly, the epitope recognized by NP137 in netrin-1 shares 90% homology with its counterpart in netrin-3, the closest member to netrin-1 in humans, for which little is known in the field of cancer. Here, we unveiled that netrin-3 appears to be expressed specifically in human neuroblastoma (NB) and small cell lung cancer (SCLC), two subtypes of neuroectodermal/neuroendocrine lineages. Netrin-3 and netrin-1 expression are mutually exclusive, and the former is driven by the MYCN oncogene in NB, and the ASCL-1 or NeuroD1 transcription factors in SCLC. Netrin-3 expression is correlated with disease stage, aggressiveness, and overall survival in NB. Mechanistically, we confirmed the high affinity of netrin-3 for netrin-1 receptors and we demonstrated that netrin-3 genetic silencing or interference using NP137, delayed tumor engraftment, and reduced tumor growth in animal models. Altogether, these data support the targeting of netrin-3 in NB and SCLC.

**Keywords** axon guidance; netrin-1; netrin-3; neuroblastoma; small cell lung cancer

**Subject Categories** Cancer; Signal Transduction

## Introduction

Navigation cues such as semaphorins, netrins, or ephrins have been shown to play various crucial roles in many cellular processes (Stoeckli, 2018). Members of the netrin family, strongly implicated in the nervous system development, have been associated with several pathologies (van Gils *et al*, 2012; Ramkhelawon *et al*, 2014; Renders *et al*, 2021). Interestingly, netrin-1, the most studied member of the netrin family, is upregulated in a large fraction of human cancers, where it is generally assumed to promote cancer cell survival *via* its interaction with its receptors, deleted in colorectal carcinoma (DCC) and members of the uncoordinated-5 family (UNC5-A, B, C, D; Mehlen *et al*, 2011; Paradisi *et al*, 2013). An effort to assay the clinical relevance of inhibiting netrin-1/receptor interactions is currently ongoing with an anti-netrin-1 monoclonal antibody NP137 (Grandin *et al*, 2016). Preliminary data from the phase 1 trial performed on patients with advanced solid cancer have recently been presented, underlining an encouraging anti-tumor activity (Cassier *et al*, 2019). A phase-1b/2 trial investigating NP137 in combination with chemotherapy or/and anti-PD1 in a specific

1 Apoptosis, Cancer and Development Laboratory- Equipe labellisée 'La Ligue', LabEx DEVweCAN, Institut PLAsCAN, Centre de Recherche en Cancérologie de Lyon, INSERM U1052-CNRS UMR5286, Université de Lyon, Centre Léon Bérard, Lyon, France
2 Small Molecules for Biological Targets, Centre de Recherche en Cancérologie de Lyon, UMR INSERM 1052 – CNRS 5286 ISPB Rockefeller, Lyon, France
3 Univ Lyon, Centre Léon Bérard, Centre de Recherche en Cancérologie de Lyon, Université Claude Bernard Lyon 1, INSERM 1052, CNRS 5286, Lyon, France
4 Toray Industries, Inc., New Frontiers Research Labs, Kanagawa, Japan
5 Netris Pharma, Lyon, France
6 CNRS UMR 8126, University Paris-Sud 11, Institut Gustave Roussy, Villejuif, France
7 Centre de Recherche en Cancérologie de Lyon, Centre Léon Bérard, Lyon, France
8 Hospices Civils de Lyon, Hôpital Edouard Herriot, Service de Gastroentérologie et d'Oncologie Digestive, Lyon Cedex 03, France
9 INSERM, U830, Génétique et Biologie des Cancers, Institut Curie, Paris, France
10 Laboratory of Clinical and Experimental Pathology, Université Côte d'Azur, CHU Nice, FHU OncoAge, Pasteur Hospital, Nice, France
*Corresponding author. Tel: +33 4 78782870; E-mail: patrick.mehlen@lyon.unicancer.fr
**Corresponding author. Tel: +33 4 69856266; E-mail: benjamin.gibert@lyon.unicancer.fr
†These authors contributed equally to this work
‡These authors contributed equally to this work as senior authors

clinical indication, i.e., uterine cancer—is also ongoing (NCT02977195).

Interestingly, the epitope recognized by NP137, a 22 amino acid sequence of the netrin-1 V2 domain (Grandin *et al*, 2016), is highly conserved in chicken netrin-2, often considered to be the ortholog of human netrin-3. Hence, we sought to investigate netrin-3 expression in a tumor-related context. Netrin-3, originally described in 1999 for its putative role as a guidance molecule, is expressed in sensory ganglia during embryonic development and displays a high level of affinity for netrin-1 receptors in mice (Wang *et al*, 1999). However, little is known on its biological functions during embryonic development as knockout mice have to our knowledge not been generated, and its expression pattern in humans has to date not been described. In the present study, we show that netrin-3 is specifically upregulated in tumors associated with neuroectodermal/neuroendocrine lineages. More specifically, we demonstrate that netrin-3 expression is correlated with neuroblastoma (NB) aggressiveness and could constitute a promising prognostic marker and may be considered as a therapeutic target in small cell lung cancer (SCLC).

## Results

### Analysis of netrin-3 expression in cancer

We first attempted to identify the cancer-associated expression of netrin-3 by conducting a bioinformatics search in public cancer databases. We first focused on a dataset encompassing RNA-sequencing data from 675 cancer cell lines (Klijn *et al*, 2015) overlapping 16 cancer cell types. Compared to *netrin-1* gene expression, which is displayed by most cancers, *netrin-3* gene expression was largely represented by two specific clusters corresponding to neuroblastoma (NB) and small cell lung cancer (SCLC) (Fig 1A). Of note, the expression of netrin-3 and netrin-1 seemed to be mutually exclusive (Fig 1B). While netrin-1 was detectable (FPKM ≥ 1) in 43.3% of cell lines, netrin-3 was only detected in 4.8%, and their common expression occurred in only 0.015% of cell lines ($P = 0.037$—Fisher's exact test) (Fig 1C), arguing in favor of a selective event driving high *netrin-3* gene expression specifically in these two neuroepithelial cancer indications (Rindi *et al*, 2018).

### Netrin-3 as a prognostic marker for NB

We thus investigated more closely the expression of *netrin-3* gene in NB, which is the most common extracranial pediatric solid tumor, responsible for 15% of all childhood cancer-related deaths, and arises from the sympatho-adrenal lineage of neural crest cells (Pugh *et al*, 2013; Matthay *et al*, 2016; Mohlin *et al*, 2019; Chang *et al*, 2020). According to the INSS classification, NB patient outcome is strongly associated with tumor grade, which encompasses five stages in the case of NB, namely 1, 2, 3, 4, and 4S. We thus analyzed *netrin-3* gene expression by qRT–PCR in a panel of 181 human NB samples (Gibert *et al*, 2014). While netrin-3 mRNA was barely detectable in stages-1, -2, and -3, which are localized NB with good prognosis, its expression increased in more advanced, highly aggressive and metastatic tumors with poor outcome (Fig 2A). Indeed, stage-4 displayed higher levels of netrin-3 compared to stages-1 ($P = 0.0102$), -2 ($P = 0.1068$), or -3

($P = 0.0042$). Remarkably, stage 4S restricted to neonates and encompassing highly metastatic tumors that often spontaneously regress, displayed a low level of netrin-3 ($P = 0.0434$; stage-4 vs. 4S) (Fig 2A). NB patients were also stratified in two groups according to MYCN amplification, the lower risk group consisting of non-MYCN-amplified and localized tumors (stages-1, -2, and -3) and metastatic forms for children under 18 months (stages-4 and 4S) (Matthay *et al*, 2016). The high-risk group includes all MYCN-amplified NB and non-MYCN-amplified stage-4 tumors, for children above 18 months (Ambros *et al*, 2003). *Netrin-3* gene expression was once again correlated with poor outcome as it was significantly higher in the high-risk group ($P = 0.039$) (Fig 2B). Furthermore, elevated *netrin-3* gene expression levels were strongly correlated with poor overall survival (OS) in this cohort (median expression ranking), with an OS at 150 months of 72.5% for low netrin-3-expressing tumors and 46.6% for high netrin-3-expressing tumors ($P = 0.029$) (Fig 2C). A group of NB patients ($n = 19$) that displayed a twofold increase or more in netrin-3 expression was extrapolated (=netrin-3 very high) from this original cohort and exhibited an OS of 28.5% ($P = 0.002$) (Figs 2C and EV1A). Moreover, in the aggressive stage-4 sustained *netrin-3* gene expression was correlated with poor prognosis, potentially underlining a function for netrin-3 in NB tumor progression and aggressiveness (Fig 2D). Finally, we confirmed the data extracted from the 181 patients, using a published cohort of 498 cases of NB patients (Zhang *et al*, 2015), and further validated this gene expression with an RNAscope analysis on fixed frozen NB-tissue samples (Figs 2E and EV1B–D).

To refine the molecular characterization of netrin-3-expressing NB tumors, we performed Gene Set Enrichment Analysis (GSEA) to identify associated pathways (Subramanian *et al*, 2005). We observed that MYCN-regulated pathways were also over-activated in the netrin-3 high group ($P = 0.039$; fdqr = 0.025) (Fig 3A). The E2F signature was also over-represented among netrin-3 high NB tumors ($P ≤ 0.001$; fdqr = 0.0014), the G2/M cell hallmark correlated with genomic instability was also predominant in netrin-3 high stage four patients (Fig EV2A-B). These two signatures are correlated with high levels of mitosis and cancer aggressiveness.

As MYCN amplification is believed to be one of the most drastic events impacting patient survival (Rickman *et al*, 2018), we decided to further analyze the links between MYCN and netrin-3 in NB among the 498 patients. As such, those with an amplified MYCN presented a higher level of netrin-3 compared to non-amplified MYCN-bearing patients ($P ≤ 0.001$), suggesting a putative direct regulation by this transcription factor (Figs 3B and EV2C). Interestingly, netrin-3 did not constitute a prognostic marker in non-MYCN-amplified patients (Fig EV2D). *Netrin-3* gene expression decreases when *mycn* was silenced using specific siRNA in two NB cell lines, whereas *netrin-1* remained unaltered (Figs 3C and EV2E). MYCN regulation of *netrin-3* gene expression was further supported, as we detected an enrichment in MYCN binding sites in the *netrin-3* promoter locus after ChIP-sequencing analysis of NB cell lines (Robinson *et al*, 2011; Zeid *et al*, 2018; Upton *et al*, 2020). This was correlated with Histone H3 acetyl-Lysine 27 (H3K27ac) and monomethylated H3K4 (H3K4me1), histone modifications that are associated with active enhancer regions (Rada-Iglesias *et al*, 2011), indicating that MYCN directly contributes to increasing the expression of netrin-3 in NB (Figs 3D and EV2F). A broad domain of the active promoter-associated epigenetic modification, trimethylated

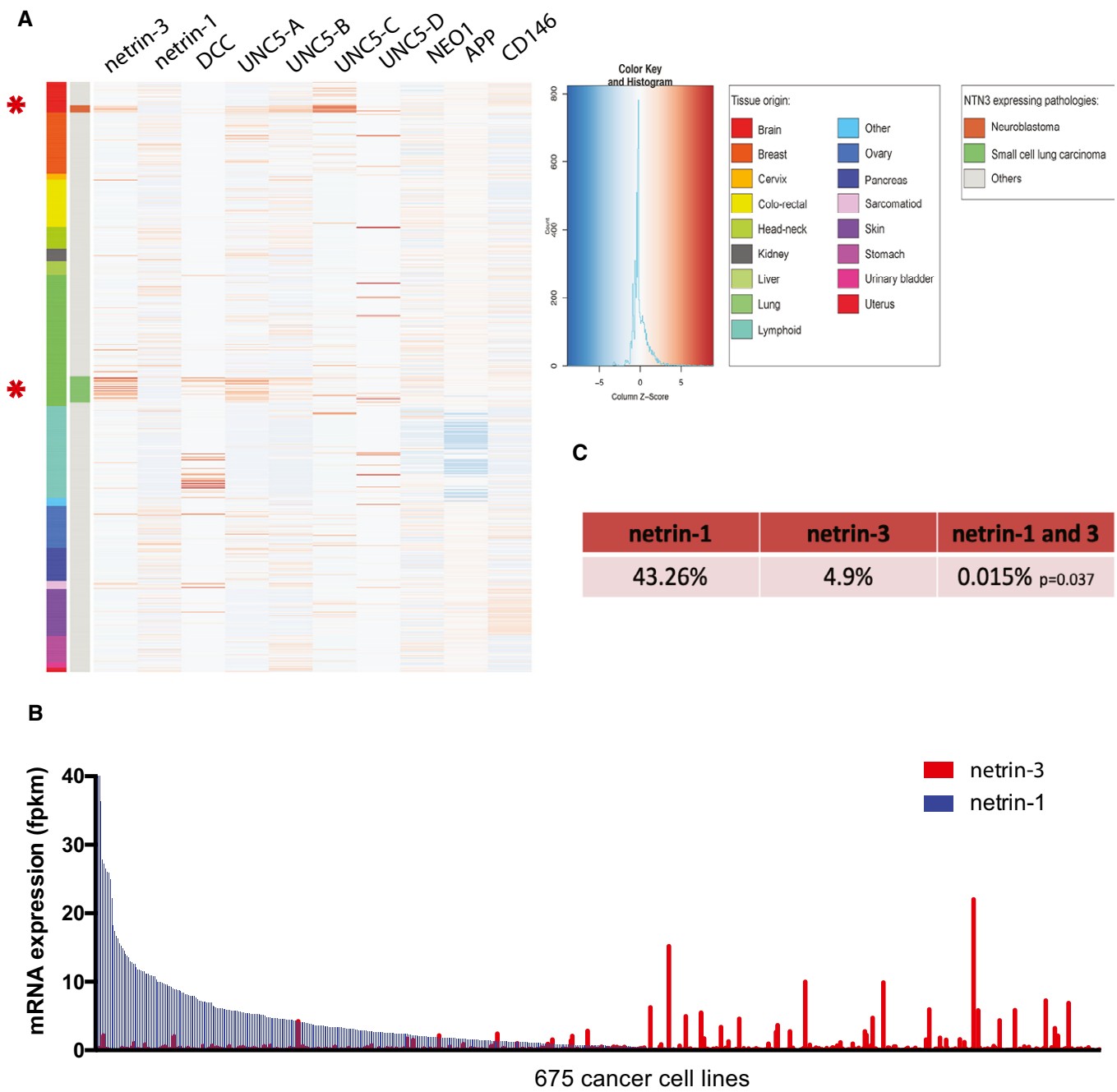

**Figure 1.  Netrin-3 expression in cancer cell lines.**

Exploring the expression by RNA sequencing of netrin-1 and netrin-3 among 675 cancer cell lines spanning 16 cancer types.

A   Expression heat map of netrin-3 and netrin-1. Receptors of netrin-1 and putative receptors of netrin-3: DCC, UNC5-A, UNC5-B, UNC5-C, UNC5-D, Neogenin, APP, CD146 are presented. The color legend (right-hand side) describes all tumor types included in the heat map. Asterisks indicate the two main pathologies in which netrin-3 expression is detectable.

B   Analysis of netrin-1 and netrin-3 expression in tumor cell lines and ranked according to netrin-1 expression.

C   Percentage of cell lines positive for netrin-1 and/or netrin-3 expression. Co-occurrence of expression was calculated using Fisher's exact test. *P*-value is indicated in the table.

H3K4 (H3K4me3), was also enriched at *netrin-3* transcription start site (TSS) and across the gene body (Figs 3D and EV2F), consistent with this being an actively transcribed gene potentially involved in

cell identity (Benayoun *et al*, 2015) (Fig EV2F). This binding was undetectable for Netrin-1 explaining, at least in part, the specificity of these expressions (Fig EV2G).

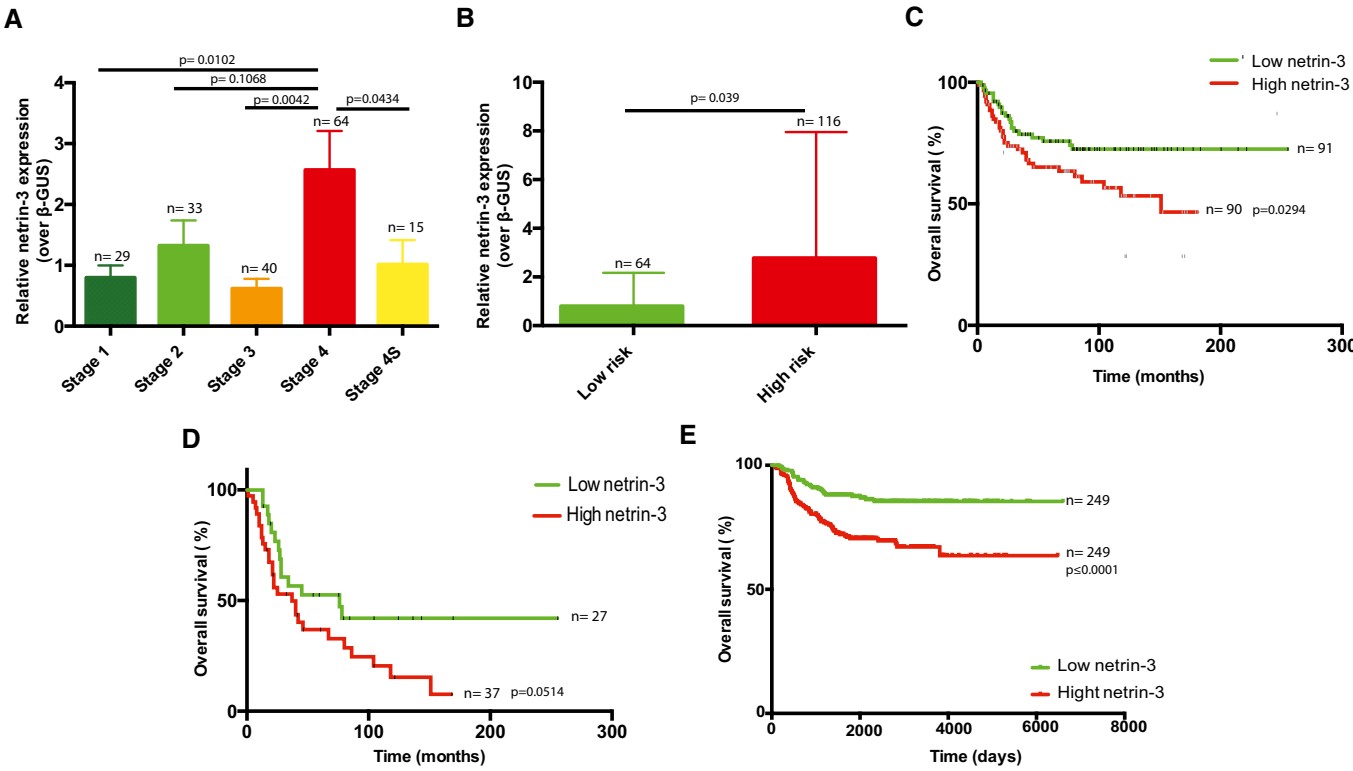

**Figure 2. Analysis of *netrin-3* expression on neuroblastoma (NB) prognosis.**

A   Quantification of *netrin-3* gene expression by qRT–PCR in a panel of 181 human NB stages 1, 2, 3, 4, and 4S. Number of cases is indicated on the graph. Error bars indicate s.e.m. Statistical treatment of the data was performed using a two-sided Student's *t*-test.

B   Quantification of *netrin-3* gene expression by qRT–PCR in a panel of 181 human neuroblastoma patients, defined as low- and high-risk NB. The number of cases is indicated on the graph. Error bars indicate s.e.m. Statistical treatment of the data was performed using a two-sided Student's *t*-test.

C   Netrin-3 high expression is a marker of poor prognosis in NB. 280 months overall Kaplan–Meier survival curves in a panel of 181 patients of all NB stages. The cohort was dichotomized base on netrin-3 median expression (in green and red). Statistical treatment of the data: Mantel–Cox; *P*-value is indicated below the graph.

D   Netrin-3 high expression is a marker of poor prognosis in aggressive NB. 280 months overall Kaplan–Meier survival curves in a panel of 64 stage four patients. The cohort was dichotomized base on netrin-3 expression, as presented in panel C. Statistical treatment of the data: Mantel–Cox; *P*-value is indicated below the graph.

E   Confirmation in 498 RNA-seq NB patients that netrin-3 high expression is a marker of poor prognosis in NB. 6,000 days overall Kaplan–Meier survival curves. The cohort was dichotomized based on netrin-3 median expression (in green and red). Statistical treatment of the data: Mantel–Cox; *P*-value is indicated below the graph.

To ascertain whether netrin-3 impacts neuroblastoma tumor formation, we conducted netrin-3 silencing *via* specific siRNA assays in the IGR-N91 cell line. The chorioallantoic membrane (CAM) of chicken embryos is a well-described model to study tumor progression (Stupack *et al*, 2006; Berthenet *et al*, 2020). NB cells were xenografted on the CAM of ten-day-old chick embryos (Fig EV2H). Seventeen-day-old chick embryos were then analyzed for primary tumor size. As shown in Fig 3E, tumors silenced for *netrin-3* were substantially smaller than controls (*U*-test; siScr vs. siNetrin-3, *P* = 0.0011). Staining of the engrafted tumors resulted in an increase in the number of apoptotic cells detected in netrin-3-silenced tumors (Cleaved PARP, *U*-test, *P* = 0.0162). It is interesting to note that proliferation appears as not modified in the same experiments (Fig EV2I). Similar results were obtained with IMR32 NB cell line (*U*-test, *P* = 0.0025) (Fig 3F).

### Elevated netrin-3 expression in small cell lung cancer

We next investigated the putative role of netrin-3 in SCLC, which is the deadliest histological subtype of lung cancers, associated with high rates of metastatic disease at diagnosis (Augert *et al*, 2020; Ko *et al*, 2021). Some drugs including immunotherapy have recently been tested in the clinic for this indication but have failed to provide benefits for a majority of patients. Netrin-3 expression was detectable by RNA sequencing in lung cancer cell lines, particularly in SCLC (*P* ≤ 0.001) and more sporadically in carcinoid and mesothelioma cells which are also neuroendocrine lineages (Figs 4A and EV3A). We performed qRT–PCR analyses to select NCI-H82 and NCI-H69 cells model, as they expressed netrin-3 but not netrin-1 (Fig 4B).

To further analyze the expression of netrin-3 in human SCLC, we performed an RNAscope analysis on FFPE sections (Fig 4C). Netrin-3 was expressed at a high level in nine out of 10 human SCLC samples tested, whereas netrin-1 was detected in only 10% of the samples, confirming our previous observation on their exclusive expression pattern (Fig 4D). Interestingly, RNAscope analysis further supported the view that netrin-3 was specifically expressed by cancer cells and not by the tumor microenvironment (Fig EV3B and C).

In an effort to unravel the mechanisms underlying the expression of netrin-3 in SCLC, we additionally carried out an *in silico*

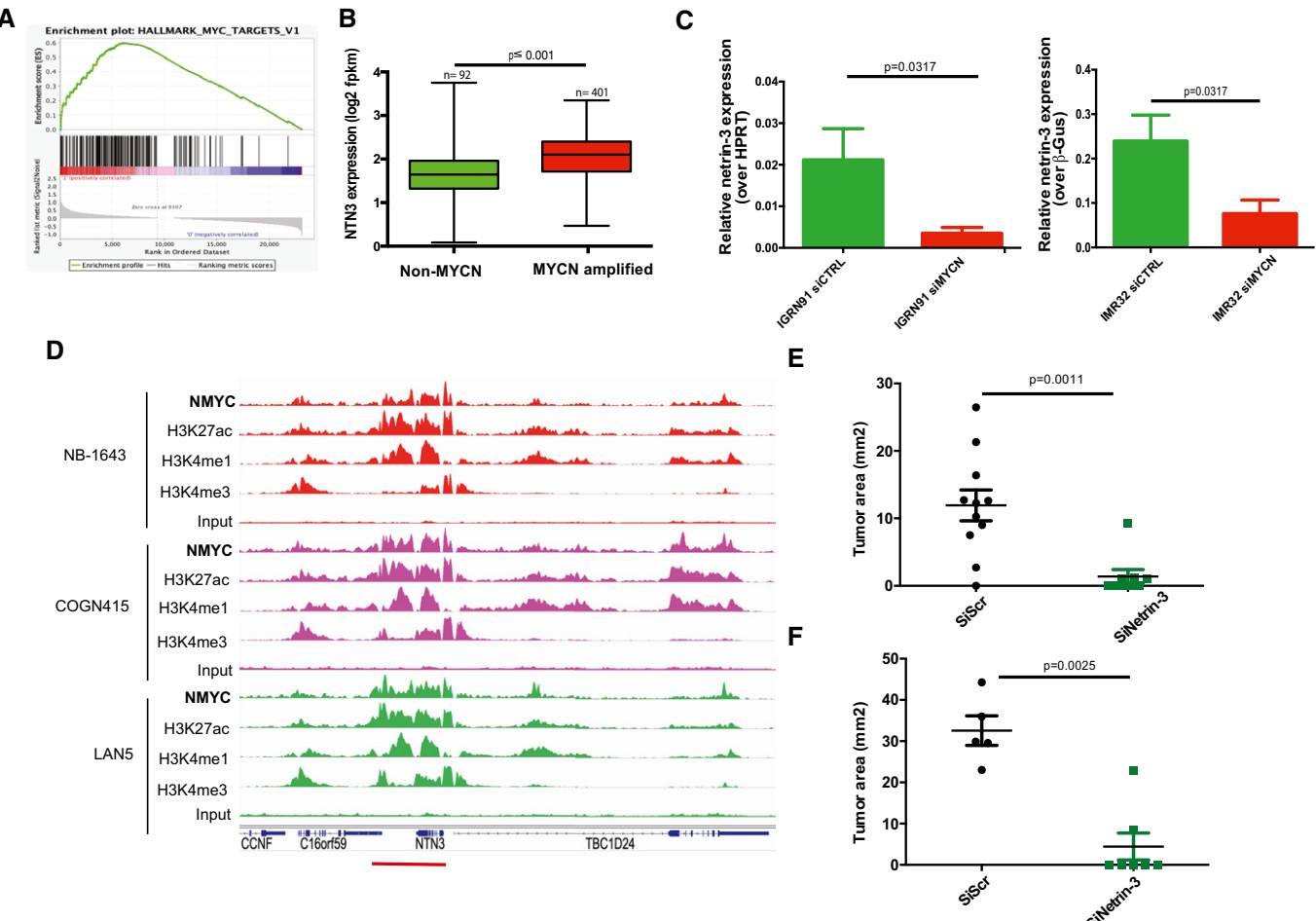

**Figure 3.   Netrin-3 is a target gene of MYCN in NB.**

A   Gene Set Enrichment Analysis (GSEA) in high netrin-3 NB patients. The cohort was dichotomized with the *n* = 50 lowest vs. 50 highest: amplification the MYCN pathway (*P* = 0.039; fdqr = 0.025).

B   Netrin-3 expression is higher in MYCN-amplified NB patients. The cohort was dichotomized based on *mycn* amplification and sorted for netrin-3 expression. Statistical treatment of the data: Welsh test; *P*-value is indicated on the graph. Error bars indicate s.e.m (boxes = 25th to 75th percentile; central band = mean; whiskers = min to max).

C   QRT–PCR analysis of *netrin-3* gene expression in IMR32 (*n* = 5) and IGRN91 (*n* = 5) NB cell lines after MYCN silencing by siRNA (*U*-test). Error bars indicate s.e.m.

D   ChIP-seq analysis of MYCN binding, active enhancer epigenetic marks H3K27ac, H3K4me1, and active promoter epigenetic mark H3K4me3, on *netrin-3* locus (red line). An enrichment of MYCN, associated with active enhancer marks, was detected in three different neuroblastoma cell lines NB-1643, COGN415, and LAN5.

E   Quantitative analysis showing the size of IGRN91 primary tumors implanted on CAM and silenced or not for netrin-3 (*n* = 11 siScr, *n* = 9 siNetrin-3; *U*-test). Error bars indicate s.e.m.

F   Quantitative analysis showing the size of IMR32 primary tumors silenced or not for netrin-3 (*n* = 5 siScr, *n* = 7 siNetrin-3; *U*-test). Error bars indicate s.e.m.

screen of transcription factors described as promoters of genes associated with SCLC aggressiveness. Indeed, netrin-3 expression is associated with the expression of two key transcription factors: the neuronal transcription factor neurogenic differentiation 1 (NeuroD1) and Achaete-Scute Family BHLH Transcription Factor-1 (ASCL-1) (Borromeo *et al*, 2016; Rudin *et al*, 2019). These factors are upregulated in a variety of aggressive neural/neuroendocrine carcinomas and important for the development and function of several neural/neuroendocrine tissues. After analysis of published ChIP-sequencing datasets, we identified that specific binding sites for both NeuroD1 and ASCL-1 are enriched upstream of the *netrin-3* gene promoter in SCLC cell lines (Robinson *et al*, 2011;

Borromeo *et al*, 2016; Huang *et al*, 2018; Upton *et al*, 2020) (Figs 5A and EV4A and B). These data show that the NeuroD1 and ASCL-1 peaks are centered around a broad domain of H3K27ac that encompasses the entire *netrin-3* upstream intergenic region, as well as most of the gene body. This suggests that NeuroD1 and ASCL-1 binding at this site is associated with transcriptional activation of netrin-3 (Fig 5B). By comparison, we did not observe a similar enrichment in H3K27ac in a wide range of other cell types, indicating selectivity to SCLC cell lines expressing NeuroD1/ASCL-1 (Fig EV4A and B). We detected no binding in the *netrin-1* gene promoter (Fig EV4C). Furthermore, *netrin-3* gene expression decreases when *neuroD1* and *ascl-1* were silenced using specific

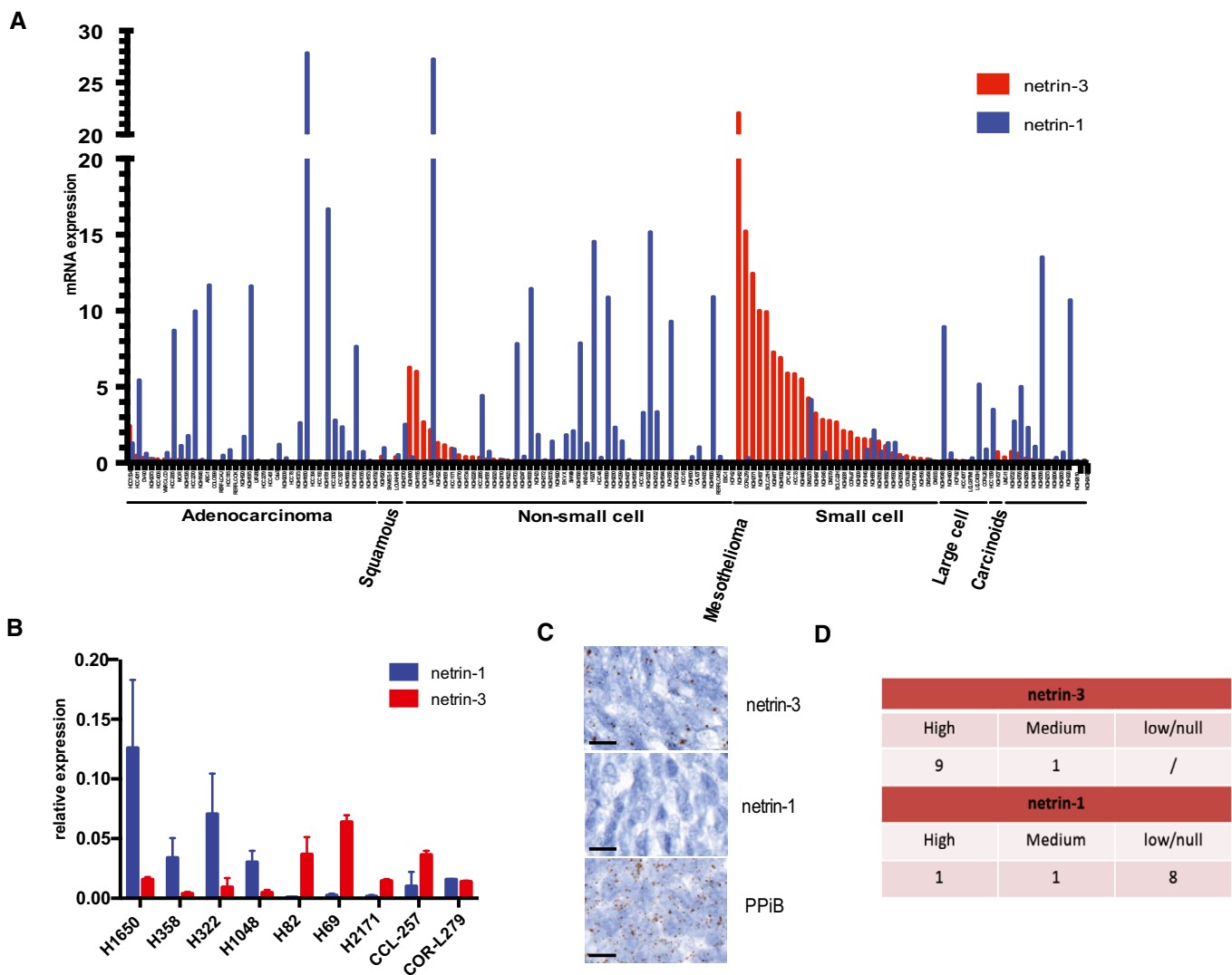

**Figure 4. Characterization of Netrin-3 in SCLC.**

A  Deep analysis of *netrin-3* gene expression in lung cancer cell lines (*n* = 76; RNA-Seq).
B  QRT–PCR analysis of netrin-1 and netrin-3 expression, in lung cancer cell Lines (Error bars indicate SD, *n* = 3).
C  Representative netrin-3 and netrin-1 mRNA detection, using RNAscope on SCLC paraffin-embedded tumor sections. Negative control DAPB, positive control PPiB. Each brown dot is a unique molecule of mRNA of each targeted gene (*n* = 10, scale bars 20 μm).
D  Quantification of netrin-3 and netrin-1 expression in human SCLC paraffin-embedded tumor sections (High ≥ 50% marked cells; Low ≤ 5%).

siRNA in two SCLC cell lines, indicating that these transcription factors directly regulate the *netrin-3* transcript (Fig 5C).

**Novel tumor growth-promoting activity by netrin-3**

Since netrin-1 has been shown to promote tumor progression in various cancer indications due to its ability to block DCC and UNC5B-induced cell death, and since our current data show that in NB and SCLC tumors, netrin-3 is upregulated rather than netrin-1, we first investigated whether netrin-3 could bind to netrin-1 receptors. Data previously reported suggested that netrin-3 could interact with netrin-1 receptors (Wang *et al*, 1999). To extend this analysis, we conducted bio-layer interferometry assays (Fig 6A). Attempts to

produce recombinant human netrin-3 were unsuccessful due to a non-soluble production of h-netrin-3 and prompted us to use chicken netrin-2L, which is phylogenetically considered as the ortholog of mammalian netrin-3. As shown in Fig 6A and B, netrin-2-like (NTN2L) was able to bind to UNC5-B and UNC5-C with a Kd similar to netrin-1, whereas it interacted with lower affinity with DCC, UNC5A, and Neogenin.

In an effort to analyze the function of netrin-3 in cancer cells, we first silenced netrin-3 and CRISPR/Cas-9 in NCI-H82 SCLC cells, using three different guide RNAs and isolated three polyclonal cell lines devoid of netrin-3 expression (Fig 6C). In this cell line and in the NCI-H82 KO polyclones, DCC was not detected. Interestingly, an increase in the level of UNC5B receptor was detected. This may

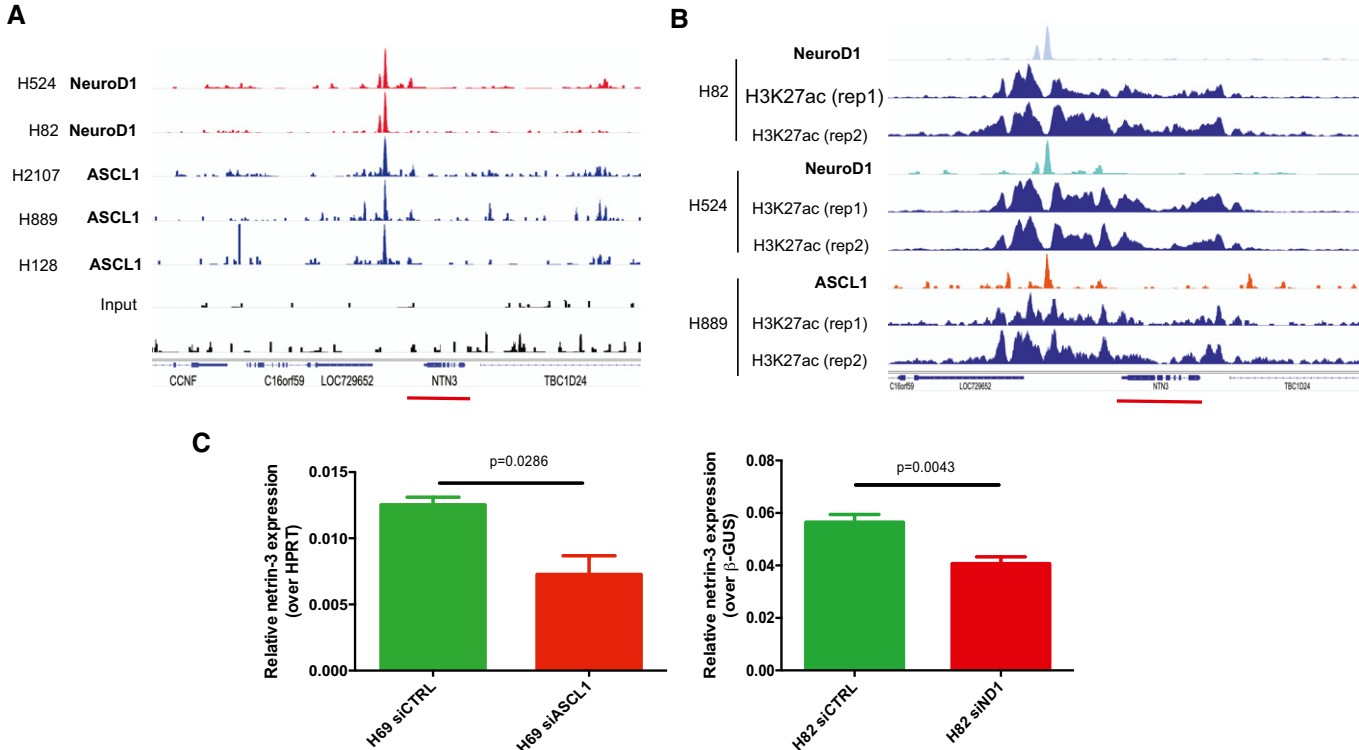

**Figure 5. Netrin-3 is a target gene of NeuroD1 and ASCL1 in SCLC.**

A  ChIP-seq analysis of NeuroD1 and ASCL-1 binding on the *netrin-3* gene promoter. A major binding site was detected (red and blue peaks for NeuroD1 and ASCL1, respectively). Input controls are shown below (black).

B  ChIP-seq analysis of NeuroD1 and ASCL-1 chromatin enrichment, which was associated with broad H3K27ac-enriched regions at the *netrin-3* locus (red line), indicating a transcriptionally active region in SCLC cell lines (rep = experimental replicates).

C  QRT–PCR analysis of *netrin-3* gene expression in NCI-H82 (*n* = 6) and NCI-H69 (*n* = 4) SCLC cell lines after Neurod-1 and ASCL-1 silencing by siRNAs (*U*-test). Error bars indicate s.e.m.

imply that UNC5B is internalized when bound to netrin-3 as formerly described for DCC when bound to netrin-1 (Bin *et al*, 2015). *In vitro*, these netrin-3 KO cells did not behave differently than control cells in terms of proliferation or cell death similarly to results obtained with siRNAs (Fig EV5A). To determine whether netrin-3 had a function in tumor formation, we engrafted these cell lines in immunodeficient mice. As shown in Fig 6D, cell populations silenced for netrin-3 expression showed a strong decrease in tumor onset, supporting a role for netrin-3 in tumor development. Similar data were observed with the NCI-H69 cell line (Fig 6E and F).

**Netrin-3 as a therapeutic target in SCLC**

Having demonstrated that netrin-3 was upregulated in two subtypes of neuroectodermal tumors, where it potentially promotes tumor development, we next sought to study its relevance as a therapeutic target. Of interest, a therapeutic anti-netrin-1 mAb, called NP137, was recently developed and is currently being tested in clinical phase 1 and 2 trials in patients harboring tumoral netrin-1 expression (Cassier *et al*, 2019). Netrin-1 and netrin-3 are 75% homologous in terms of protein conservation, and this homology reaches 90.1% for the epitope region recognized by the NP137 (Figs 7A and

EV5B). To investigate whether this similarity is sufficient to allow NP137 to bind to netrin-3, we tested the binding of NP137 on recombinant chicken netrin-2 (rchNTNT2) by bio-layer interferometry. As shown in Fig 7B, NP137 clearly bound netrin-2L within a similar range of affinity compared to netrin-1 (Kd rhNTN1: 0.96 ± 0.30 nM; Kd for rchNTN2: 1.53 ± 0.96 nM) (Figs 7B and EV5C). Moreover, using the netrin-1/UNC5B interaction as a positive control, we further showed that NP137 also inhibits the netrin-2L/UNC5B interaction in a dose-dependent manner (Fig 7C). Next, we assessed whether as observed for netrin-1-expressing cancer cells (Grandin *et al*, 2016), NP137 could affect netrin-3-expressing SCLC tumor cells. NCI-H82 cells were thus grafted in immunodeficient mice, which were then received systemic injections of NP137 once tumors had reached 100 mm³. As shown in Fig 7D, NP137 triggers a significant tumor growth inhibition and an enhanced mouse survival (Figs 7E and EV5D).

Similar data were observed with the NCI-H2286 cell line, highlighting netrin-3 as a potential therapeutic target (Fig EV5E). To decipher, the mechanism of action of the antibody targeting netrin-3 we conducted an immunohistological analysis of anti-netrin-treated NCI-H82 tumors, which revealed a significant increase of cleaved PARP staining (Figs 7F and EV5F). Interestingly, no significant modification of Ki67 is detected confirming chorioallantoic

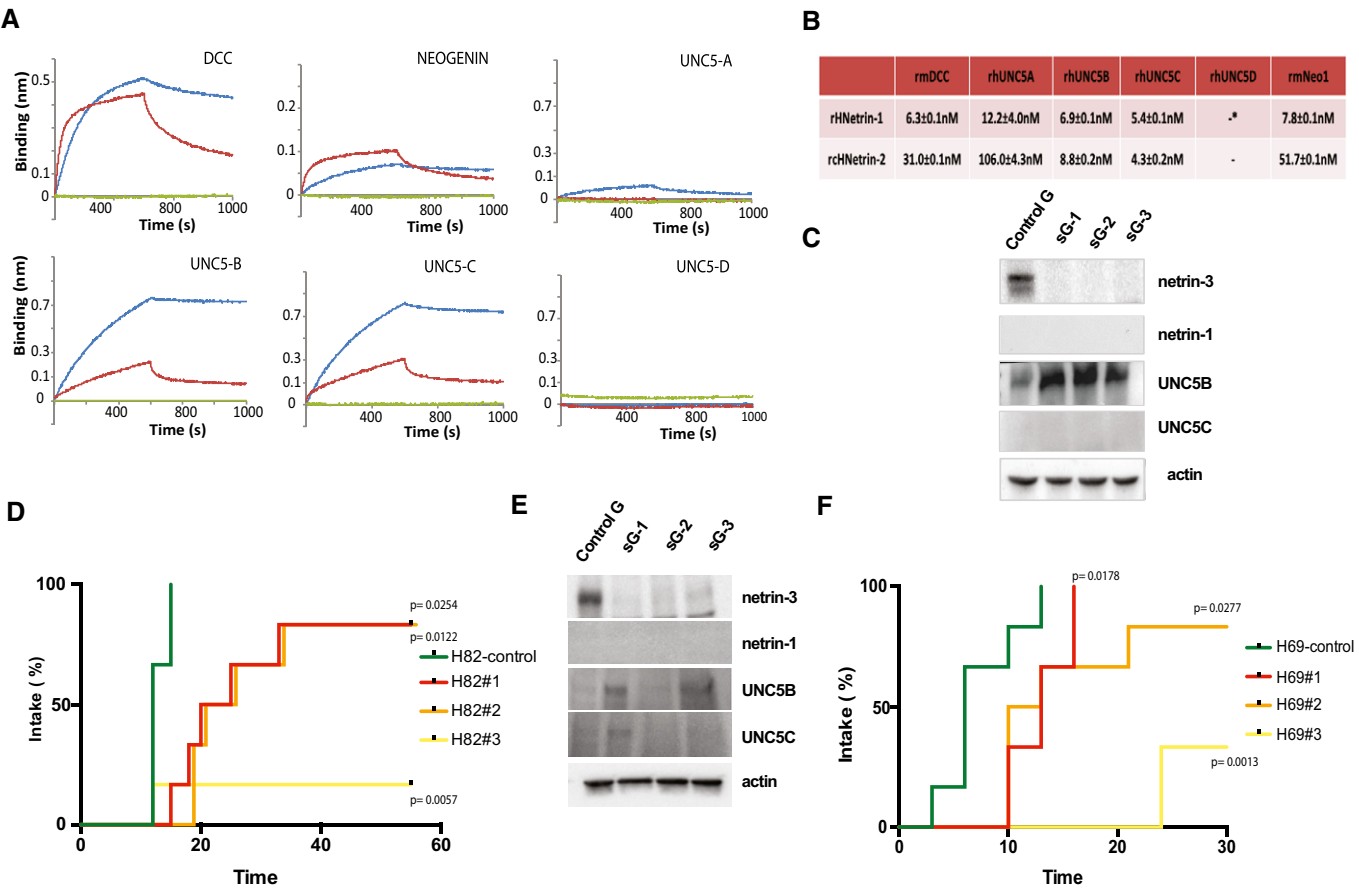

**Figure 6. Tumor promoting role of netrin-3.**

A   Bio-layer interferometry analysis of netrin-2L/receptor interactions. Other members of the netrin family were also tested: netrin-1, netrin-4 and netrin-G2 (red: netrin-2L; blue: netrin-1; green netrin-4).

B   Kd calculation after bio-layer interferometry analysis, *= no interaction detected.

C   NCI-H82 CRISPR/Cas9 were subjected to immunoblots. sGx, designates the number of guide RNAs, control cells (sGc).

D   Netrin-3 silencing delays or inhibits SCLC engraftment *in vivo*. NMRI *nude* mice were subcutaneously engrafted with either NCI-H82-CRIPSR/cas9-NTN3 with three different guides RNAs (sG1 P = 0.0022; sG2 P ≤ 0.001; and sG3 P ≤ 0.001) or control cells (sGc) n = 6/group. Positive tumor engraftment was reported when tumors reached 50 mm$^3$ (Mantel–Cox).

E   NCI-H69-CRISPR/Cas9 were subjected to immunoblots. sGx designates the number of guide RNAs, control cells (sGc).

F   NCI-H69-CRISPR/Cas9 cells presented in E. were subcutaneously engrafted with either NCI-H69- CRIPSR/cas9-NTN3 with three different guide RNAs (sG1 P = 0.017; sG2 P = 0.028; and sG3 P = 0.001) or control cells (sGc) n = 6/group. Positive tumor engraftment was reported when tumors reached 20 mm$^3$ (Mantel–Cox).

Source data are available online for this figure.

---

membrane experiments, further supporting the view that netrin-3 interference is *in vivo* associated with cancer cell death.

## Discussion

In the present paper, we analyzed the expression and possible functions of netrin-3, an understudied protein in the field of oncology. We show that netrin-3 is selectively expressed in two neuroectodermal human cancer types, NB and SCLC. Its high level of expression and strong specificity to these two cancers, support a possible role for netrin-3 in these pathologies. Indeed, netrin-3 is highly expressed in high-grade compared to low-grade NBs, implying a strong correlation between netrin-3 expression and patient survival

at pathological stages and in all aggressive tumor types. Netrin-3 could thus emerge as a diagnostic marker, and owing to its secreted nature, this may provide promising opportunities for its routine detection in patient blood samples. Nevertheless, and in order to further assess netrin-3 levels in pre-clinical/clinical studies, a specific netrin-3 antibody is necessary for its immunohistochemical characterization, as we failed to validate the commercially available candidates we tested.

Importantly, we demonstrated that netrin-3 is also expressed in SCLC tumor cells, at least in part through the activities of ASCL1 and Neuro-D1 (Borromeo *et al*, 2016). Analyses of published ChIP-sequencing datasets of these transcription factors revealed that they are abundant on chromatin upstream of the *netrin-3* gene, which is correlated with high levels of epigenetic marks associated with

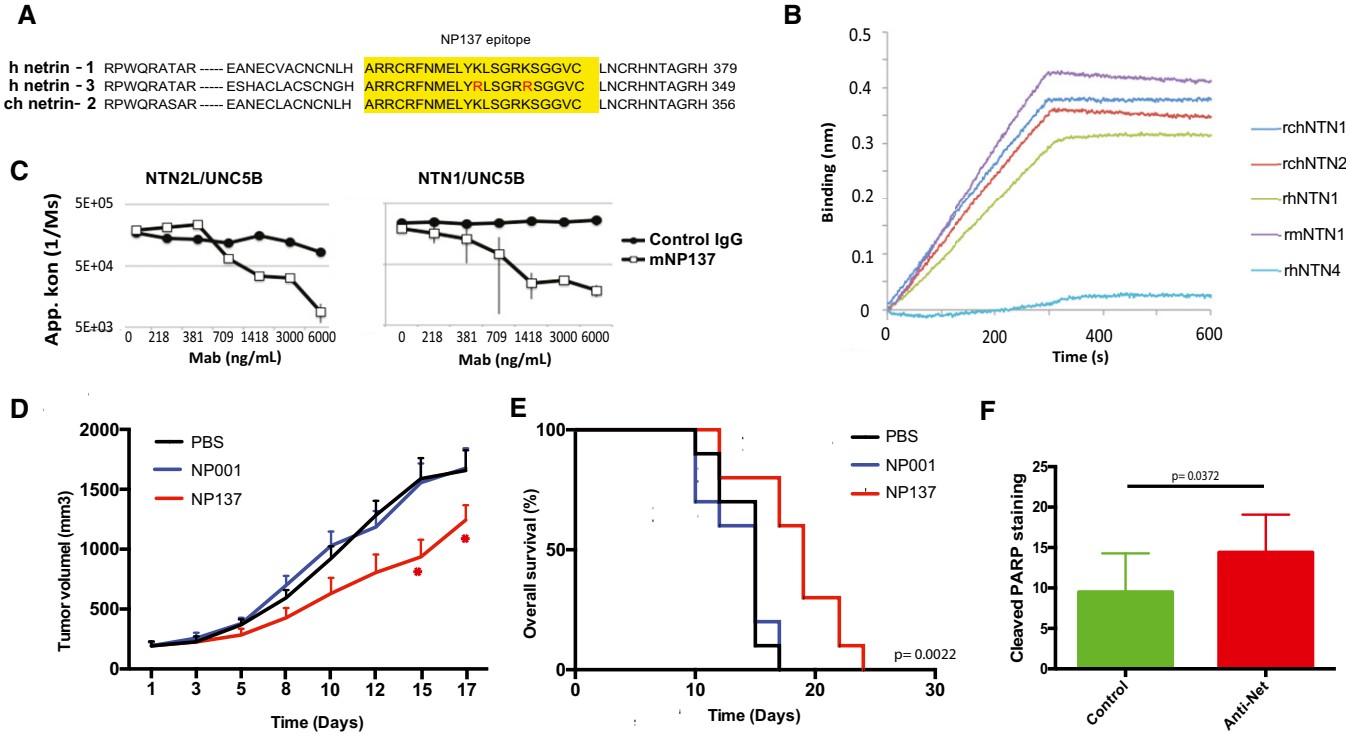

**Figure 7. Netrin-3 is a therapeutic target in SCLC.**

A  Analysis of the NP137 antibody (anti-netrin-1) epitope across netrin family members.

B  Analysis of NP137 binding on netrin-2L, netrin-1; netrin-4 or netrin-G2 by bio-layer interferometry assays. Kd rhNTN1: 0.96 ± 0.30 nM; Kd for rchNTN2: 1.53 ± 0.96 nM.

C  NP137 inhibits NTN1/UNC5B or NTN-2L/UNC5B interaction, detected by Elisa assays. Here the recombinant proteins are the murine forms (Control IgG = mouse IgG2A, mNP137 = murine Fc, UNCB = rat UN5H2) bars indicate SD, n = 3.

D  NMRI *nude* mice were engrafted with NCI-H82 cells by subcutaneous injection of 2 million cells. When the mean tumor volume reached approximately 80 mm$^3$, animals were treated 3 × time/weekly by intra-abdominal injection of PBS; NP001 (Human IgG1, isotype); NP137 (anti-netrin1/3) during 20 days. n = 10 animals/group (*P < 0.05, two-way ANOVA, error bars indicate s.e.m.).

E  NP137 enhances survival of mice engrafted with NCI-H82 cell line (see D). Analysis of Kaplan–Meier survival curves of mice treated or not with NP137. Mantel–Cox test; n = 10 animals/group.

F  Quantification of cell death (cleaved PARP) in H82-treated tumors in D. Tumors were analyzed 7 days after the first treatment (*i.e.*, 400 mm$^3$; n = 3/group. U-test, bars indicate SD).

active enhancers (H3K27ac and, when data available, H3K4me1). Netrin-3 was also expressed in NB, and we determined *in vitro* that the driver of NB, namely MYCN, was bound upstream of the netrin-3 gene. Similarly to ASCL1 and Neuro-D1 in ASCL, this was correlated with high levels of active enhancer epigenetic marks (H3K27ac, H3K4me1). H3K4me3 formed a peak at the *netrin-3* promoter in MYCN-positive NB cell lines, supporting the view that *netrin-3* is a constitutively transcribed gene in these cells, potentially associated with cell identity (Benayoun *et al*, 2015).

Small cell lung cancer is generally regarded as highly aggressive and treatment options remain limited (Polley *et al*, 2016; Antonia *et al*, 2016). Our present data using genetic silencing or protein titration support the view that targeting netrin-3 may offer promising perspectives for SCLC patients. Of interest, there is currently a candidate drug tested in clinics with excellent safety data (Cassier *et al*, 2019), that we show here to interact with netrin-3 with an affinity in the nanomolar range by inhibiting netrin-3/receptor interactions. We present evidence that inhibiting netrin-3/receptor

interactions with NP137 is associated with tumor growth inhibition in SCLC pre-clinical models. Future works should further explore whether NP137 could be an attractive treatment for SCLC as monotherapy or in combination with chemotherapies and immunotherapies.

Our study raised several puzzling issues, including the apparent mutually exclusive expression of netrin-1 and netrin-3, and the restricted expression of netrin-3 to NB and SCLC, while netrin-1 expression is displayed by most cancer indications. In terms of biological evolution, these proteins have conserved a strong structural homology, and our data together with previous findings suggest that netrin-3 binds to the same receptors as netrin-1, thus functionally mimicking netrin-1 in particular in NB (Delloye-Bourgeois *et al*, 2009). Preliminary data from our laboratory consistently show that netrin-3 can block cell death induced by the dependence receptor UNC5B. However, both expression in tumors and during embryonic development seems to be completely different. *Netrin-1* silencing in mice is associated with embryonic lethality and major

nervous system defects (Bin *et al*, 2015), whereas netrin-3 expression appears to be restricted to the peripheral nervous system (Wang *et al*, 1999). Yet, the specific expression of netrin-3 in neuroepithelial tumors, which may indicate lineage specificity, may also provide important tools in terms of diagnosis. Future work should investigate whether blood/urine detection of netrin-3 could represent an original predictor of SCLC, for which there is a clear clinical unmet need.

# Materials and Methods

### Human tumor samples

Patients or their legal guardians signed a written informed consent agreeing on the use of tumor samples for research, according to the French regulations on the protection of persons (French Ethics Committee). Human NB samples were collected from a cohort of 181 patients staged in accordance with the International Neuroblastoma Staging System (Gibert *et al*, 2014). Material was collected by the Institut Gustave Roussy (Villejuif, France). Human SCLC tumors were collected by the Biobank Tissue unit, Pasteur Hospital (Nice, France). Informed consent was obtained from all subjects, and the experiments conformed to the principles set out in the WMA Declaration of Helsinki and the Department of Health and Human Services Belmont Report. Following patient agreement, primary tumors were obtained, either by biopsy or after surgery, and were directly frozen before storage.

### Mouse experiments and pathology

Animals were maintained in a specific pathogen-free animal facility (Anican, Lyon—France) and stored in sterilized filter-topped cages. Mice were handled in agreement with the institutional recommendations and procedures approved by the animal care committee (Comité d'Evaluation Commun au Centre Léon Bérard, à l'Animalerie de transit de l'ENS, au PBES et au laboratoire P4; CECCAP). Seven-week-old (20–22 g body weight) female NMRI mice were obtained from Janvier Labs (Saint Berthevin—France) (Delloye-Bourgeois *et al*, 2013). The animals were housed in standard cages (five animals/cage) with litter and food. Enrichment was achieved by adding cotton for nesting.

### Generation of stable netrin-3 knocked out cell line

Different small guide RNAs (sgRNA) were designed using the sgRNA designer tool (CRISPRi), (https://portals.broadinstitute.org/gpp/public/analysis-tools/sgrna-design-crisprai). The sequences are as follows: sg1: 5'-CCGCCCTCGCTGCAGCCGGG-3'; sg2: 5'-CGCCCAC GGCCCTTCCCGGG-3'; sg3: 5'-TCGCTGCAGCCGGGAGGAGG-3'.

The Esp3I linker was added to sgRNAs, and the oligonucleotides were then annealed by incubation for 3 min at 90°C and 15 min at 37°C and ligated in pLV hU6-sgRNA hUbC-dCas9-KRAB-T2a-Puro (Addgene #71236) (Thakore *et al*, 2015). Insertion of sgRNA was confirmed by sequencing. The use of pLV hU6-sgRNA hUbC-dCas9-KRAB-T2a-Puro, which is constructed around a 3$^{rd}$ generation lentiviral backbone, allows for the simultaneous expression of Cas9 and gRNA and for selection for puromycin resistance. Resulting

plasmids were used to produce lentiviruses to further infect cell lines of interest. Lentiviral vectors were a generous gift from Carine Maisse (INRA, UMR754, Lyon, France).

### Lentiviral transduction

Production of VSV-g pseudotyped lentivirus was made by co-transfecting 70–80% confluent 293FT cells (gift from Dr Fabrice Lavial, CRCL) with lentiviral vectors described in 1) and 2) plus pCMV-dR8.91 and pCMV-VSV-g plasmids using Lipofectamine 3000 in OptiMEM medium + GlutaMAX supplemented with 1 mM Sodium Pyruvate and 5% FBS. Transfection medium was replaced 6 h later by OptiMEM + Pyruvate + 5% FBS. Lentiviral supernatant was harvested 48 h later, filtered through a 0.45-µm syringe filter, and used either fresh or snap-frozen. For transduction, cell lines of interest were plated in 0.25 ml in 24-well plates and 0.25 ml of viral supernatant was added in the presence of 10 µg/ml polybrene. 24 h later, viral supernatant was discarded and cells were cultured in their normal culture medium supplemented with puromycin for one week.

### Cell culture

NCI-H82 and NCI-H69 were obtained from ATCC and cultured in RPMI 1640 Medium (ATCC 30-2001) adding fetal bovine serum (FBS, ATCC 30-2020) to a final concentration of 10% and 1% penicillin/streptomycin (P/S). NCI-H2286 was obtained from ATCC and cultured in DMEM: F12 Medium (Catalog No.30-2006) adding 0.005 mg/ml Insulin, 0.01 mg/ml Transferrin, 30 nM Sodium selenite (final concentration), 5% FBS and 1% P/S. HeLa cells were obtained from our lab and cultured in DMEM with 10% FBS and 1% P/S. All cells were tested for mycoplasma contamination.

### Quantitative RT–PCR

Tumor cells, patient samples, or xenografts were disrupted with MagNALyser kit (Roche Applied Science). Total RNAs of either cell lines or tumors were extracted by using NucleoSpin® RNAII Kit (Macherey Nagel) according to the manufacturer's protocol. RT–PCR reactions were performed with iScript cDNA Synthesis Kit (BIO-RAD). One microgram total RNA was reverse transcribed by using the following program: 25°C for 5 min, 42°C for 30 min, and 85°C for 5 min. Real-time quantitative RT–PCR was performed with a LightCycler 96 apparatus (Roche) using LightCycler® TaqMan® Master kit (Roche, Basel, Switzerland). Expression of target genes was normalized to hypoxanthine-guanine phosphoribosyltransferase (HPRT), glyceraldehyde-3-phosphate dehydrogenase (GAPDH), and beta-glucuronidase (GUSB) using the comparative CT method.

Relative expression of each gene was calculated according to the comparative $2-\Delta\Delta Ct$ quantification method in which $\Delta Ct = Ct$ (Sample)$-Ct$ (Normalizer) and $\Delta\Delta Ct = \Delta Ct$ (Sample)$-\Delta Ct$ (Calibrator).

hASCL1 F (probe 38) CGACTTCACCAACTGGTTCTG; hASCL1 R (probe 38) ATGCAGGTTGTGCGATCA; hMYCN F (probe 38) TAATATGCCCGGGGGGACT; hMYCN R (probe 38) GGGCTGGAA CTGGCTTTT; hNeuroD1 F (probe 30) CTGCTCAAGGACCTACTAA CAACAA; hNeuroD1 R (probe 30) GTCCAGCTTGGAGGACCTT; hNTN3 F (probe 56) GCCCTGTGTTAAGACCCCTA, hNTN3 RR (probe 56) TGCAGTGCGAGTCACAGTC; hNTN1 F (probe 3) AAAA GTACTGCAAGAAGGACTATGC; hNTN1 R (probe 3) CCCTGCTTA

TACACGGAGATG; hHPRT F (probe 73) TGACCTTGATTTATTTTG CATACC; hHPRT R (probe 73) CGAGCAAGACGTTCAGTCCT; hGUS F (probe 57) CGCCCTGCCTATCTGTATTC; hGUS R (probe 57) TCCCCACAGGGAGTGTGTAG.

### Bio-layer interferometry

BLI experiments were performed using The OctetRED96 system. All interactions were analyzed at 30°C with constant shaking at 1,000 rpm in PBS, 0.02% Tween-20, 0.1% BSA (binding buffer: BB). To determine the affinity between hNetrin-1 or chNetrin-2 with the extra-cellular domain (ECD) of hUNC5H1, rUNC5H2, hUNC5H3, and hUNC5H4, netrin-1 or netrin-2-coated-HIS1K biosensors were incubated with an increasing concentration series of ECDs (3.12 nM, 6.25 nM, 12.5 nM, 25 nM, 50 nM, 100 nM) and binding was observed for 5 min at 30°C. Biosensors were then incubated in BB for a further 5 min to observe dissociation of the complex. To determine the affinity between h-netrin-1 or ch-netrin-2 with the extra-cellular domain (ECD) of mNeogenin and mDCC, ECDs-coated-AHC biosensors were incubated with a concentration series of hNetrin-1 or chNetrin-2 (3.12 nM, 6.25 nM, 12.5 nM, 25 nM, 50 nM, 100 nM). In this assay, 1 μg/ml Dextran 5,000 was added to BB to prevent non-specific binding of the netrins to the biosensors. Association and dissociation were monitored as described above. To determine the affinity of the human anti-netrin NP137 antibody for h-netrin-1 and ch-netrin2, Netrin-1 or Netrin-2-coated-HIS1K biosensors were incubated with an increasing concentration series of NP137 (1 nM, 2 nM, 5 nM, 10 nM, 20 nM, 50 nM, 100 nM) and association was observed for 5 min at 30°C. Biosensors were then incubated in BB for a further 5 min to observe dissociation of the complex. To determine the ability of the anti-netrin antibody to antagonize the binding of h-netrin-1 and ch-netrin-2 with UNC5H2, the ECD of UNC5H2 was captured on AHC biosensors, incubated with 25 nM of netrin-1 or netrin-2L in presence of various concentrations of either the murine anti-netrin antibody or an isotype control (64 pM, 320 pM, 1.6 nM, 8 nM, 40 nM). The association was observed for 5 min at 30°C to evaluate kon. Binding kinetics were evaluated with ForteBio Octet RED Evaluation software 6.1 using a 1:1 binding model to derive kon, koff, and KD values.

### Western blot

Cells were harvested and lysed in SDS buffer (5% SDS, 10% Glycerol, 10 mM Tris–HCl pH 7.6, 1% Triton X-100, 0.1 M DTT) and then sonicated using a Branson Digital sonifier 450 for 20 pulses (50% amplitude, pulse on 0.2 s, pulse off 1.0 s). Protein concentration was measured with the 660 nm Protein assay kit (Pierce Biotechnology, Rockford, IL, USA) using bovine serum albumin (BSA) as a standard curve according to manufacturer's instructions. Protein extracts (20–50 mg per lane) were loaded onto 4%–15% SDS-polyacrylamide gels (Bio-Rad) and transferred onto nitrocellulose membranes using Trans-Blot Turbo Transfer System (Bio-Rad). Membranes were blocked with 5% skimmed milk in PBS/0.1% Tween-20 (PBS-T) for 1 h and then incubated overnight with primary antibody: 1:1,000 dilution for anti-netrin-1 (Abcam, anti-NTN1 ab126729), 1:500 dilution for anti-netrin-3 Abcam (anti-NTN3 ab185200),

After three washes with PBS-T, membranes were incubated with the appropriate HRP-conjugated secondary (1:5000 dilution)

antibody for 1 h at room temperature. Detection was performed using West Dura Chemiluminescence System (Pierce). Membranes were imaged on the ChemiDoc Touch Imaging System (Bio-Rad).

### Human histopathology

For histological examination, tissue samples were fixed in 10% buffered formalin and embedded in paraffin (FFPE) (see Supplementary Materials section). 4-μm-thick FFPE tissue sections were prepared according to conventional procedures. Sections were then stained with hematoxylin/eosin and examined under a light microscope. mRNA ISH was performed using the RNAscope 2.5 VS Reagent kit—BROWN with a custom designed netrin-3 probe (Advanced Cell Diagnostic, Hayward, CA) according to the manufacturer's guidelines. The RNAscope procedure was performed using the Discovery XT autostainer with mRNA amplification, pretreatment & DAB PTO kit (Roche, Meylan, France). Tissue control was assessed by performing RNAscope analysis of common housekeeping genes PPIB and DAPB, as a negative control. Finally, sections were scanned with panoramic scan II (3DHistech, Budapest, Hungary) at 40× and Z-stacked (3 sections at 0.6 μm intervals).

10-μm-thick cryosections were obtained from tissues and stored at −80°C. Frozen tissue slides were immersed in 4% formalin for 1 h and then rinsed in PBS. Tissues were dehydrated in increasing percentages of alcohol and air-dried.

The mRNA ISH was performed using the RNAscope 2.5 VS reagent kit brown with custom designed netrin-3 probes (Advanced Cell Diagnostic, Hayward, CA) according to the manufacturer's instructions. The RNAscope procedure was performed in the Discovery XT autostainer with mRNA amplification, pretreatment and DAB PTO kit (Roche, Meylan, France), without steps of deparaffinization and HIER. Tissue control was assessed by performing RNAscope analysis of a common housekeeping gene PPiB. Finally, sections were scanned with panoramic scan II (3DHistech, Budapest, Hungary) at 40× and Zstack (3 levels spaced of 0.6 μm).

### ChIP-seq analysis

NeuroD1 ChIP-seq peaks in NCI-H524 and NCI-H82 cells, ASCL1 ChIP-seq peaks in H2107, NCI-H889, and NCI-H128 cells, and the corresponding input controls for NCI-H524 and NCI-H889 cells, were downloaded from NCBI GEO dataset GSE69398 (PMID: 27452466) and GSM1700637; GSM894066; GSM894072; GSM894100; GSM89409; GSM1700641; GSM1526706; GSM1526702 and analyzed using IGV software. NeuroD1 and ASCL1 ChIP-seq data were downloaded from NCBI GEO dataset GSE69398 (PMID: 27452466). All H3K27ac datasets were downloaded from GSE115123 (PMID: 29945888). Data were analyzed using IGV software.

All neuroblastoma ChIP-seq tracks were downloaded from NCBI GEO dataset GSE138315 (bioRxiv 829754; https://doi.org/10.1101/829754), GSM2113521, GSM2127461, GSM1680108. Data analyzed using IGV software.

### Cell death and proliferation assays

Apoptosis was monitored by measuring caspase-3 activity as described previously [18] using Caspase 3/CPP32 Fluorometric

**The paper explained**

**Problem**
Small cell lung cancer (SCLC) is the deadliest histological subtype of lung cancers, associated with high rates of metastatic disease at diagnosis. Some drugs including immunotherapy have recently been tested in the clinic for this indication but have failed to provide benefits for a majority of patients.

**Results**
The tumoral expression of netrin-3 is restricted to SCLC and neuroblastoma (NB). Surprisingly, netrin-1 and netrin-3 have mutually exclusive expressions. Netrin-3 is correlated with patient overall survival and poor prognosis in human NB, which is associated with the MYCN oncogene amplification. Expression of netrin-3 in SCLC is proposed to be at least in part controlled by the ASCL-1 and NeuroD1 transcription factors. In addition, systemic administration of the NP137 monoclonal antibody, originally designed against netrin-1, decreases tumor growth in animal models.

**Impact**
Netrin-3 was targeted by the anti-netrin-1 antibody, which is already used in the clinic, and could thus represent new therapeutic opportunities.

Assay Kit (Gentaur Biovision, Brussel, Belgium) 24 h after transfection with siRNAs.

For proliferation assays, $10^5$ cells were seeded onto a 96-well plate. After 24 h, 20 μl of CellTiter96®AQueous One Solution Reagent was added to each well containing 100 μl of culture medium. The plate was then incubated at 37°C for 2 h in a humidified, 5% $CO_2$ atmosphere. Absorbance at 490 nm was recorded using a TECAN-infinite m1000.

**CAM, mouse experiments, and pathology**

The chorioallantoic membrane (CAM) of chicken embryos is a well-described model to study tumor progression (Stupack *et al*, 2006; Berthenet *et al*, 2020). The fertilized eggs were obtained from a producer (EARL Les Bruyères, Dangers, France). They were kept in specific incubators. IGR-N91 and IMR32 cells (15 and 10 million) were xenografted on the CAM of ten-day-old chick embryos with Matrigel (100 μl/tumor) at day 10. Seventeen-day-old chick embryos were then analyzed for primary tumor size as previously described (Gibert *et al*, 2014).

For the engraftment experiments, mice were implanted with either NCI-H82 or NCI-H69 SCLC cells; control or CRISPR-netrin-3 #1, #2, #3, by subcutaneous injection of $10^6$ cells in 100 μl of PBS (and 100 μl of Matrigel) into the right flank of mice. Tumor volume was calculated with the formula V = (length*width2)/2. A positive tumor catch is declared when the tumor reaches 50 mm$^3$ for NCI-H82 and 20 mm$^3$ for NCI-H69.

Mice were implanted with either NCI-H82 or NCI-H69 SCLC cells and NCI-H2286 mix cell line, by subcutaneous injection of $10^6$ cells in 100 μl of PBS into the right flank. Mice were treated with either PBS, NP001 (isotype control human IgG1) or NP137 (anti-Netrin-1/3, human IgG1) by *i.v.* injection at 10 mg/kg twice a week during 3 weeks, once tumors were established at a volume close to 100 mm$^3$ after external quantification using calipers. Tumor volume was calculated with the formula V = (length*width$^2$)/2. At the end of

the treatment, tumors were harvested and embedded in paraffin, then sectioned into 10-μm slices. Tumor histology was studied after hematoxylin-Phloxine B-saffron, Ki67, cleaved caspase-3, and cleaved PARP staining's of tumor slides at the Pathology-Research Platform (Lyon-France).

**Statistical analyses**

Quantitative reverse-transcriptase PCR (qRT–PCR) and RNA-sequencing data analyses were performed using the GraphPad Prism (San Diego, California) and R software (free software foundation, University of Auckland, New Zealand) to generate heat maps. Unpaired Student's *t*-test with Welch correction was used to compare the different groups. Cohorts were separated based on median netrin-3 expression. Survival curves were produced according to the Kaplan–Meier method on the GraphPad software. Data were analyzed with a Mantel–Cox test. *n* defines the total replicates. All statistical tests were two-sided.

# Data availability

This study includes no data deposited in external repositories.

**Expanded View** for this article is available online.

# Acknowledgements

We thank Brigitte Manship for proofreading the manuscript and Robert Dante and Arnaud Augert for helpful discussions. This work was supported by institutional grants from University of Lyon (PM), Centre Léon Bérard (PM), INSERM (PM), and CNRS (PM, BG). This work was also supported by grants from Fondation ARC for young investigators (BG), and from the Ligue Contre le Cancer (PM), INCA (PM). SJ was supported by a LabEx DEVweCAN fellowship, and MR was supported by la Ligue Contre le Cancer fellowship.

# Author contributions

SJ, MR, SB, MH, BD, JGD, and ARR performed the cell experiments and *in vitro* data. SJ, MSa, MR, and DN performed mice experiments. MR, MH, and SB performed CAM experiments. AM and DG produced the lentiviruses and CRISPR cell lines; NG made anatomopathological analysis; PV, MSi, and PMu performed methylation data and promoter analysis. NR, PS, TW, BG, and SOC performed bioinformatical analysis; PH, IJL, VC, and CHG provided human samples and cell lines. PMe and BG designed the research and wrote the paper.

# Conflict of interest

BD, DN, DG, and PM declare to have a conflict of interest as respectively employees (BD, DN, and DG) and shareholders (PM) of Netris Pharma.

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
