## [Review Process File · EMBO Molecular Medicine]

Targeting netrin-3 in Small Cell Lung Cancer and Neuroblastoma.

Shan Jiang, Mathieu Richaud, Pauline Vieugué, Nicolas Rama, Jean Guy Delcros, Maha Siouda³, Mitsuaki Sanada, Anna-Rita Redavid, Benjamin Ducarouge, Maeva Hervieu, Silvia Breusa, Ambroise Manceau, Charles-Henry Gattolliat, Nicolas Gadot, Valérie Combaret, David Neves, Sandra Ortiz-Cuaran, Pierre Saintigny, Olivier Meurette, Thomas Walter, Isabelle Janoueix-Lerosey, Paul Hofman, Peter Mulligan, David Goldshneider, Patrick Mehlen and Benjamin Gibert
DOI: 10.15252/emmm.202012878

Corresponding authors: Patrick Mehlen (mehlen@lyon.fnclcc.fr) , Benjamin Gibert (benjamin.gibert@lyon.unicancer.fr)

Review Timeline:

Submission Date:	4th Jun 20
Editorial Decision:	7th Jul 20
Revision Received:	18th Dec 20
Editorial Decision:	15th Jan 21
Revision Received:	1st Feb 21
Accepted:	7th Feb 21

Editor: Lise Roth

Transaction Report:

7th Jul 2020

Dear Dr. Mehlen,

Thank you for submitting your work to EMBO Molecular Medicine. We have now heard back from the three referees who agreed to evaluate your manuscript. As you will see below, the reviewers raise substantial concerns on your work, which unfortunately preclude its publication in EMBO Molecular Medicine in its current form.

The reviewers find the question addressed by the study of interest, however they remain unconvinced that some of the major conclusions are sufficiently supported by the data. In particular, the following points should be addressed:

- validation of the results in additional in vivo models,
- further mechanistic understanding.

If you feel you can satisfactorily address these points as well as the other points listed by the referees, you may wish to submit a revised version of your manuscript.

Addressing the reviewers' concerns in full will be necessary for further considering the manuscript in our journal, and acceptance of the manuscript will entail a second round of review. EMBO Molecular Medicine encourages a single round of revision only and therefore, acceptance or rejection of the manuscript will depend on the completeness of your responses included in the next, final version of the manuscript. For this reason, and to save you from any frustrations in the end, I would strongly advise against returning an incomplete revision.

When submitting your revised manuscript, please carefully review the instructions that follow below. Failure to include requested items will delay the evaluation of your revision:

- 1) A .docx formatted version of the manuscript text (including legends for main figures, EV figures and tables). Please make sure that the changes are highlighted to be clearly visible.
- 2) Individual production quality figure files as .eps, .tif, .jpg (one file per figure).
- 3) A .docx formatted letter INCLUDING the reviewers' reports and your detailed point-by-point responses to their comments. As part of the EMBO Press transparent editorial process, the point-by-point response is part of the Review Process File (RPF), which will be published alongside your paper.
- 4) A complete author checklist, which you can download from our author guidelines (<https://www.embopress.org/page/journal/17574684/authorguide#submissionofrevisions>). Please insert information in the checklist that is also reflected in the manuscript. The completed author checklist will also be part of the RPF.
- 5) Before submitting your revision, primary datasets produced in this study need to be deposited in an appropriate public database (see <https://www.embopress.org/page/journal/17574684/authorguide#dataavailability>).

Please remember to provide a reviewer password if the datasets are not yet public. The accession numbers and database should be listed in a formal "Data Availability " section (placed after Materials & Method). Please note that the Data Availability Section is restricted to new primary data that are part of this study.

6) We would also encourage you to include the source data for figure panels that show essential data. Numerical data should be provided as individual .xls or .csv files (including a tab describing the data). For blots or microscopy, uncropped images should be submitted (using a zip archive if multiple images need to be supplied for one panel). Additional information on source data and instruction on how to label the files are available at .

7) Our journal encourages inclusion of *data citations in the reference list* to directly cite datasets that were re-used and obtained from public databases. Data citations in the article text are distinct from normal bibliographical citations and should directly link to the database records from which the data can be accessed. In the main text, data citations are formatted as follows: "Data ref: Smith et al, 2001" or "Data ref: NCBI Sequence Read Archive PRJNA342805, 2017". In the Reference list, data citations must be labeled with "[DATASET]". A data reference must provide the database name, accession number/identifiers and a resolvable link to the landing page from which the data can be accessed at the end of the reference. Further instructions are available at .

8) We replaced Supplementary Information with Expanded View (EV) Figures and Tables that are collapsible/expandable online. A maximum of 5 EV Figures can be typeset. EV Figures should be cited as 'Figure EV1, Figure EV2' etc... in the text and their respective legends should be included in the main text after the legends of regular figures.

- Additional Tables/Datasets should be labeled and referred to as Table EV1, Dataset EV1, etc. Legends have to be provided in a separate tab in case of .xls files. Alternatively, the legend can be supplied as a separate text file (README) and zipped together with the Table/Dataset file. See detailed instructions here: .

9) The paper explained: EMBO Molecular Medicine articles are accompanied by a summary of the articles to emphasize the major findings in the paper and their medical implications for the non-specialist reader. Please provide a draft summary of your article highlighting

10) For more information: There is space at the end of each article to list relevant web links for

further consultation by our readers. Could you identify some relevant ones and provide such information as well? Some examples are patient associations, relevant databases, OMIM/proteins/genes links, author's websites, etc...

11) Author contributions: the contribution of every author must be detailed in a separate section (before the acknowledgments).

12) A Conflict of Interest statement should be provided in the main text

13) Every published paper now includes a 'Synopsis' to further enhance discoverability. Synopses are displayed on the journal webpage and are freely accessible to all readers. They include a short stand first (maximum of 300 characters, including space) as well as 2-5 one-sentences bullet points that summarizes the paper. Please write the bullet points to summarize the key NEW findings. They should be designed to be complementary to the abstract - i.e. not repeat the same text. We encourage inclusion of key acronyms and quantitative information (maximum of 30 words / bullet point). Please use the passive voice. Please attach these in a separate file or send them by email, we will incorporate them accordingly.

Please also suggest a striking image or visual abstract to illustrate your article. If you do please provide a png file 550 px-wide x 400-px high.

14) As part of the EMBO Publications transparent editorial process initiative (see our Editorial at <http://embomolmed.embopress.org/content/2/9/329>), EMBO Molecular Medicine will publish online a Review Process File (RPF) to accompany accepted manuscripts.

In the event of acceptance, this file will be published in conjunction with your paper and will include the anonymous referee reports, your point-by-point response and all pertinent correspondence relating to the manuscript. Let us know whether you agree with the publication of the RPF and as here, if you want to remove or not any figures from it prior to publication.

I look forward to receiving your revised manuscript.

Yours sincerely,

Lise Roth

Lise Roth, PhD
Editor
EMBO Molecular Medicine

To submit your manuscript , please follow this link:

Link Not Available

Photos 400-800 DPI

*Additional important information regarding figures and illustrations can be found at <http://bit.ly/EMBOPressFigurePreparationGuideline>

***** Reviewer's comments *****

Referee #1 (Remarks for Author):

This is an excellent paper demonstrating tissue specific overexpression of netrin-3 and its potential therapeutic effect in neuroectodermal tumors, including neuroblastoma (NB) and small cell lung cancer (SCLC). The study is well organized with a lot of convincing results of netrin-3 expression in human material screening data and clinical data, the extensive molecular analyses of transcriptional regulation of netrin-3 in NB and SCLC and growth promoting activity in immunodeficient mice, and the experimental therapeutics of SCLC tumors in immunodeficient mice by a humanized monoclonal antibody against netrin-1 and netrin-3. Considering the secretory features of netrin-3 together with a newly identified specific receptor, UNN5B, this cascade would provide a promising novel diagnostic marker and a therapeutic target for NB and SCLC, both of which needs additional diagnostic and therapeutic approaches to improve the prognosis of patients. Especially, the findings of the significant suppression of tumor size of human SCLC cells in immunodeficient mice and the resultant improvement of overall survival of the mice in experimental therapeutics by a monoclonal antibody, NP137, are promising with high impact to the audience.

For the benefit of the readers, the authors are recommended to answer the following comments and questions.

Major comments:

1. The authors are strongly recommended to introduce the physiological and developmental function of Netrin-3 and the phenotype in gene deficient mice a little bit more in detail, in addition to a brief introduction of its expression in sensory ganglion in Page 3.
2. Figure 4 is very interesting and suggestive to speculate the significance of netrin-3. In addition to NB and SCLC, we can see that netrin-3 is expressed in at least 2 carcinoid cell lines, NCI-H727 and UMC-11, 2 cell lines from asbestosis-derived mesothelioma, NCI-H2722 and NCI-H2795. In addition,

NCI-H1156 listed in NSCLC in the left hand, is shown to produce neuromedin B, which might suggest partially neuronal origin. Please include some description about netrin-3 expression in carcinoid and mesothelioma cells.

3. Figure 6A: Molecular interaction between immunoglobulin superfamily (IgSF) molecules is often difficult due to their relatively weak affinity and this reviewer believes that the bio-layer interferometry is a critical approach to detect specific and quantitative interactions. Please include detailed description of the principle and methods of the bio-layer interferometry analysis in Materials and Methods as well as the reason why the authors used it in the text for the benefit of readers.

Minor comments:

4. Page 5, the 2nd paragraph: Over-presentation of E2F signature and dysregulation of pRB in netrin-3 high NB tumors is interesting. Is netrin-3 expression also found in retinoblastoma, another tumor of neuronal origin that lacks RB suppressor function? Please send a comment.

5. Page 6, line 3 from the bottom: NeuroD1 and ASCL-1 are known to act as transcription factors involved in SCLC of neuroendocrine features, which corresponds to more than 70% of SCLC. Please include the percentage of SCLC that express netrin-3 and the description that NeuroD1 and ASCL-1 are transcription factors involved in SCLC of neuroendocrine features for the benefit of readers.

6. Page 7, line 12: Please explain possible reasons why recombinant human netrin-3 is difficult to produce.

7. Page 7, line 19: Chicken NTN2L is able to bind UNC-5B and 5C. Then, this reviewer wants to know the expression of UNC5C in addition to UNC5B in NCI-H82 cells and H82 netrin-3 KO cells. Please describe it in the text.

8. NCI-H series of small cell lung cancer cells are recommended to describe their full names, like NCI-H82, instead of H82 as described in Figure 4.

Referee #2 (Remarks for Author):

The authors have performed a series of experiments to evaluate the role of Netrin-3 in the neuroectodermal tumors neuroblastoma and small-cell lung cancer (SCLC). The manuscript presents data on the relative expression of Netrin-1 and Netrin-3 and the associations of Netrin-3 gene expression with outcomes and prognostic features in both neuroblastoma and SCLC. The investigators also explore potential mechanisms of Netrin-3 gene regulation and the efficacy of Netrin-3 depletion and targeting with an anti-Netrin-1 antibody NP137 in SCLC models.

The manuscript is generally well written, although the grammar and overall writing could use significant editing by a native English speaker. The manuscript has a few gaps and flaws that should be addressed as detailed below. The exploration of the efficacy of targeting Netrin-3 is also somewhat superficial. The key concerns and other issues that need to be addressed are:

1) The authors refer to Netrin-3 expression throughout the manuscript but should clarify that they are referring to NTN3 gene expression, rather than Netrin-3 protein expression

2) Although the authors state NTN1 and NTN3 are mutually exclusive, but there are cell lines and tumors that appear to express both - analyses of these cell lines or other models with dual expression would potentially be useful. Furthermore, documenting exclusive (or shared) protein expression would be more functionally relevant

3) All of the analyses performed rely on NTN3 gene expression, even in the patient tumor samples.

Some measure of relative protein expression would be much more functionally relevant

4) In the "Netrin-3 as a prognostic marker for NB" section, the authors state that NB patients are stratified into two groups "according to MYCN expression" which is incorrect - this statement should be corrected

5) The Kaplan-Meier curve in Figure 2C should include curves for both tumors with very high NTN3 expression and with those high NTN3 expression (with the very-high NTN3 expression excluded)

6) The authors mention the E2F gene signature identified by GSEA - were there other gene signatures identified in the analysis?

7) the authors' descriptions of the relative roles of E2F and MYCN in neuroblastoma is very confusing - MYCN amplification is an independent prognostic factor, leading to increased MYCN expression. There is no evidence that MYCN amplification is linked to E2F expression or function. Are the authors saying that E2F drives MYCN amplification? Or MYCN expression (in either MYCN-amplified or MYCN-nonamplified tumors)? Or are E2F and MYCN independently regulating NTN3 expression? The statement that "MYCN directly contributes to increase the expression of netrin-3 in NB" has not been proven and should be corrected (factually and grammatically).

8) The authors should consider including data on NTN1 from the ChIP-seq results presented in Figures 3 and 5 to show whether NTN3 and NTN1 share regulatory features

9) The authors refer to Figure "2H" at the end of the section on NB - there is no such figure provided

10) In the section on SCLC, the authors state that "netrin-3 expression is associated with the expression of two key transcription factors.." - this statement has not been proven and should be corrected

11) In the section "Novel tumor growth promoting activity by netrin-3" the authors state that netrin-3 KO cells did not have differences in cell proliferation or death "(data not shown)" - this data could easily be shown in a supplemental figure

12) validation of the results in SCLC cells and tumors with NTN3 knockdown and NP137 in neuroblastoma models would dramatically enhance the impact of the manuscript

13) the authors only include images from single tumor samples in Figures 4B and Suppl Fig 1A - additional images from more samples (such as a tissue microarray) would be much more powerful, as Figure 4B in particular is not very convincing

14) the impact of NP137 on SCLC xenograft tumors does not appear to be very dramatic - how do these effects compare to responses in NTN1-expressing tumors? The authors should consider including a comment on potential relative efficacy/response rates

15) how are the "Intake (%)" numbers calculated in Figure 6D and 6F?

16) the specific patient datasets used for Suppl Fig 2A,B are not clear and should be included in the figure (or figure legend)

Referee #3 (Comments on Novelty/Model System for Author):

As detailed in the remarks to the authors, I feel that the in vivo experiments somewhat lack a low number of cell lines an inconsistent analysis between the netrin-3 deletion experiments and the intervention study with the netrin-1 antibody.

Referee #3 (Remarks for Author):

The manuscript by Jiang et al. investigates the potential role of netrin-3 in tumours. The authors find selected expression in two neuroectodermal tumour types, neuroblastoma (NB) and small cell lung cancer (SCLC). Expression of netrin-3 appears to be mutually exclusive with netrin-1 which has a broader expression pattern in cancer, and targeting of which is currently investigated in clinical trials.

Jiang et al. first establish the expression pattern of netrin-3 in neuroblastoma and its potential use as a prognostic marker in NB. Additionally, they briefly touch on potential mechanisms for transcriptional regulation of netrin-3 in NB and SCLC, mainly via ChIPSeq experiments. In the third part of the manuscript, the feasibility of targeting netrin-3 in SCLC is investigated. Using CRISPR/Cas9 deletion of netrin-3 or using an antibody against netrin-3 that cross-reacts with netrin-3 the authors demonstrate that netrin-3 plays a role in tumor-growth and/or initiation. In principle, the study provides some interesting data and targeting netrin-3 might provide an interesting therapeutic option. However, I feel the current manuscript touches upon several points without investigating each in sufficient depth to completely convince the reader.

Main points:

After the extensive analysis of netrin-3 in NB one would have expected that in vivo experiments targeting netrin-3 would also be performed with NB cell lines. This could add to the importance of netrin-3 expression in NB and the general applicability of netrin-3 targeting in cancer.

In general, the number of cell lines used for the in vivo studies are quite low. A more extensive follow-up analysis of the in vivo experiments should be performed. Histology (H&E) with stainings for e.g. Ki-67 could be informative.

What is the suggested mechanism of action? For the experiments with netrin-3 deletion, the authors write that 'tumour development' is impaired. However, the experiments measure tumour take-rate which would be an indicator of clonogenicity. Why didn't the authors measure tumour growth in these experiments?

Are there any molecular mechanisms the authors could propose? They state that neither proliferation nor apoptosis were changed in the netrin-3 deleted cells. It might be beneficial to show those important data, and also provide at least some idea what the underlying mechanism for the impaired in vivo growth is.

The analysis of transcription factor binding sites in the netrin-3 gene are somewhat correlative. A knockdown or knockout of the respective transcription factors would easily provide more meaningful data on the regulation of netrin-3 expression.

Additional points:

Figure 2E: The analysis of the validation cohort should also be performed separately for stage-4

patients (as in Fig. 4D).

Figure 3: The GSEA appears to be performed on the whole cohort. As the authors demonstrate, netrin-3 expression correlates with stage and MYCN status the results are somewhat expected. An additional GSEA only in stage-4 netrin-3 high vs. low patients might provide more interesting results.

The legend to Figure 4 refers to a non-existing subfigure (q-RT-PCR)

Figure 6A: The figure is missing a legend for the colour keys in the affinity experiments.

Fig. 6E: What is the netrin receptor expression on H69 cells?

The experiments for affinity measurements and the production of recombinant ligands are not described in the methods section.

Dear editor,

Thank you for the positive feedback and interest in our manuscript. We greatly appreciate the referees' constructive comments that have enabled us to improve the quality of our manuscript entitled "*Targeting Netrin-3 in Small Cell Lung Cancer and Neuroblastoma*". We have endeavored to address the comments made by the three referees and we believe that the resulting revision is now more compelling.

Referee #1 (Remarks for Author):

This is an excellent paper demonstrating tissue specific overexpression of netrin-3 and its potential therapeutic effect in neuroectodermal tumors, including neuroblastoma (NB) and small cell lung cancer (SCLC).

The study is well organized with a lot of convincing results of netrin-3 expression in human material screening data and clinical data, the extensive molecular analyses of transcriptional regulation of netrin-3 in NB and SCLC and growth promoting activity in immunodeficient mice, and the experimental therapeutics of SCLC tumors in immunodeficient mice by a humanized monoclonal antibody against netrin-1 and netrin-3. Considering the secretory features of netrin-3 together with a newly identified specific receptor, UNC5B, this cascade would provide a promising novel diagnostic marker and a therapeutic target for NB and SCLC, both of which needs additional diagnostic and therapeutic approaches to improve the prognosis of patients. Especially, the findings of the significant suppression of tumor size of human SCLC cells in immunodeficient mice and the resultant improvement of overall survival of the mice in

experimental therapeutics by a monoclonal antibody, NP137, are promising with high impact to the audience.

We thank Reviewer 1 for her/his kind comments on our study and we hope we will respond to all of his/her concerns.

For the benefit of the readers, the authors are recommended to answer the following comments and questions.

Major comments:

1. The authors are strongly recommended to introduce the physiological and developmental function of Netrin-3 and the phenotype in gene deficient mice a little bit more in detail, in addition to a brief introduction of its expression in sensory ganglion in Page 3.

As suggest by reviewer 1, this point has been clarified and a more general introduction on Netrin-3 expression during embryonic development was presented. To our knowledge Netrin-3 knock out mice have never been generated and published, but this work will for sure be of great interest to understand the role of Netrin-3 in both physiological conditions and pathologies.

2. Figure 4 is very interesting and suggestive to speculate the significance of netrin-3. In addition to NB and SCLC, we can see that netrin-3 is expressed in at least 2 carcinoid cell lines, NCI-H727 and UMC-11, 2 cell lines from asbestosis-derived mesothelioma, NCI-H2722 and NCI-H2795. In addition, NCI-H1156 listed in NSCLC in the left hand, is shown to produce neuromedin B, which might suggest partially neuronal origin. Please include some description about netrin-3 expression in carcinoid and mesothelioma cells.

Referee one is right when he points out that Netrin-3 could sporadically be expressed by other cell and tumor types. The text has been modified accordingly.

3. Figure 6A: Molecular interaction between immunoglobulin superfamily (IgSF) molecules is often difficult due to their relatively weak affinity and this reviewer believes that the bio-layer interferometry is a critical approach to detect specific and quantitative interactions. Please include detailed description of the principle and methods of the bio-layer interferometry analysis in Materials and Methods as well as the reason why the authors used it in the text for the benefit of readers.

As highlighted by referee one, this important point has been added in the material and methods with a new section on bio-layer interferometry.

Minor comments:

4. Page 5, the 2nd paragraph: Over-presentation of E2F signature and dysregulation of pRB in netrin-3 high NB tumors is interesting. Is netrin-3 expression also found in retinoblastoma, another tumor of neuronal origin that lacks RB suppressor function? Please send a comment.

Retinoblastoma is certainly a pathology of great interest that we have never studied. As suggested in the comment of Reviewer 1, we have analyzed the expression of netrin-3 in

retinoblastoma cohorts available in a public database (Irsan E Kooi *et al.*, 2015). As a result, and as suggested by Reviewer 1, netrin-3 is detectable in retinoblastoma apparently at low levels. As shown in the present preliminary data, no differences are detectable between retinoblastoma subgroups defined in the study we referred to (Irsan E Kooi *et al.*, 2015), which does not suggest a specific mechanism. Nevertheless, we will try to understand why netrin-3 may be important in retinoblastoma in future analyses.

5. Page 6, line 3 from the bottom: NeuroD1 and ASCL-1 are known to act as transcription factors involved in SCLC of neuroendocrine features, which corresponds to more than 70% of SCLC. Please include the percentage of SCLC that express netrin-3 and the description that NeuroD1 and ASCL-1 are transcription factors involved in SCLC of neuroendocrine features for the benefit of readers.

New comments were addressed in the revised form to explain in details the regulation of Netrin-3 by ASCL-1 and NeuroD1. A new figure 5C has been added on the regulation of Netrin-3 by ASCL-1 and NeuroD1 showing a more direct regulation than described in the first version.

6. Page 7, line 12: Please explain possible reasons why recombinant human netrin-3 is difficult to produce.

This point was addressed in the revised form of the manuscript.

7. Page 7, line 19: Chicken NTN2L is able to bind UNC-5B and 5C. Then, this reviewer wants to know the expression of UNC5C in addition to UNC5B in NCI-H82 cells and H82 netrin-3 KO cells. Please describe it in the text.

This has been changed in the revised form. Western blot assays were performed in SCLC cell lines to characterize all netrin receptors. As a result, the Unc5C receptor is not expressed in this cell line, indicating that the effect we detected are specifically mediated by UNC5B (Figure 6E).

8. NCI-H series of small cell lung cancer cells are recommended to describe their full names, like NCI-H82, instead of H82 as described in Figure 4.

This has been amended in the revised form.

Referee #2 (Remarks for Author):

The authors have performed a series of experiments to evaluate the role of Netrin-3 in the neuroectodermal tumors neuroblastoma and small-cell lung cancer (SCLC). The manuscript presents data on the relative expression of Netrin-1 and Netrin-3 and the associations of Netrin-3 gene expression with outcomes and prognostic features in both neuroblastoma and SCLC. The investigators also explore potential mechanisms of Netrin-3 gene regulation and the efficacy of Netrin-3 depletion and targeting with an anti-Netrin-1 antibody NP137 in SCLC models.

The manuscript is generally well written, although the grammar and overall writing could use significant editing by a native English speaker. The manuscript has a few gaps and flaws that should be addressed as detailed below. The exploration of the efficacy of targeting Netrin-3 is also somewhat superficial.

We thank reviewer 2 for his/her kind comments on our manuscript. As suggested, we had the manuscript proofread and edited by a native English speaker, Brigitte Manship, who kindly corrected the grammatical errors in our text.

The key concerns and other issues that need to be addressed are:

1) The authors refer to Netrin-3 expression throughout the manuscript but should clarify that they are referring to NTN3 gene expression, rather than Netrin-3 protein expression
This point has been modified in the text in the revised version of the manuscript.

2) Although the authors state NTN1 and NTN3 are mutually exclusive, but there are cell lines and tumors that appear to express both - analyses of these cell lines or other models with dual expression would potentially be useful. Furthermore, documenting exclusive (or shared) protein expression would be more functionally relevant.

As shown in Figure 1C only 0.015% of 675 cell lines express both Netrin-1 and Netrin-3. This point is crucial to understand if Netrin proteins could be functionally redundant. To determine this specific point, we characterized the only cell line we found by Western blot co-expressing Netrin-1 and Netrin-3: the Neuroblastoma IMR32 cell line (Expanded View 2I). To complete the characterization of Netrins in cells lines, a new q-RT-PCR-analysis has been conducted in lung cancer cell lines and is now presented in figure 4B, but we failed to detect any co-expressing cell line.

The expression of Netrin-1 and -3 within human tumor samples is more difficult to address as Netrin-1 is also expressed by other cell types like cancer associated fibroblasts and immune cell populations. We performed an analysis on the Human TCGA database but as SCLC and NB are not available, we did not find representative Netrin-3 expression among other pathologies.

3) All of the analyses performed rely on NTN3 gene expression, even in the patient tumor

samples. Some measure of relative protein expression would be much more functionally relevant

We fully agree that this point is important for the conclusions of our article. As you know, Neuroblastoma is a tumor of the very young child where only biopsies are now taken, and small cell lung cancer is never operated on (unless misdiagnosed). Due to this total unavailability of samples, we have not been able to perform a Western blot on patient samples.

In order to find a solution, we undertook an anatomopathology analysis on patient samples fixed in paraffin. Unfortunately, we were not able to find an antibody that could function properly in immunohistochemistry. We therefore tested the 6 commercially available antibodies for Netrin-3 and obtained a strong background in human samples that did not express the protein at the messenger level. These data on the protein are therefore, in our opinion, not compatible with a publication, which is why we have developed the RNAscope strategy. We have added a comment to this effect in the discussion of this limitation in our study.

In order to continue our study on Netrin-3 in the future, we would like to develop such an antibody in order to better describe its role (rate, localization...) in human samples.

4) In the "Netrin-3 as a prognostic marker for NB" section, the authors state that NB patients are stratified into two groups "according to MYCN expression" which is incorrect - this statement should be corrected

This sentence has been modified in the revised manuscript. The correct sentence is now: according to MYCN amplification.

5) The Kaplan-Meier curve in Figure 2C should include curves for both tumors with very high NTN3 expression and with those high NTN3 expression (with the very-high NTN3 expression excluded)

This point was modified in the new version of the manuscript. A figure based on netrin-3 expression median was done, and a new one has been performed accordingly to reviewer recommendation (Figure 2C, Expanded View 1C).

6) The authors mention the E2F gene signature identified by GSEA - were there other gene signatures identified in the analysis?

We found only two signatures in NTN3-high *versus* NTN3-low patients by GSEA analysis. As mentioned in the article, only the MYCN oncogene amplification signature and E2F transcription

factors were activated in patients classified as NTN3-high.

7) the authors' descriptions of the relative roles of E2F and MYCN in neuroblastoma is very confusing - MYCN amplification is an independent prognostic factor, leading to increased MYCN expression. There is no evidence that MYCN amplification is linked to E2F expression or function. Are the authors saying that E2F drives MYCN amplification? Or MYCN expression (in either MYCN-amplified or MYCN-nonamplified tumors)? Or are E2F and MYCN independently regulating NTN3 expression? The statement that "MYCN directly contributes to increase the

expression of netrin-3 in NB" has not been proven and should be corrected (factually and grammatically).

We apologize for the lack of clarity of our explanations. As mentioned in point N°5, we have only identified by GSEA two modulated signatures in neuroblastoma patients with a high rate of expression of netrin-3, which are MYCN and E2F. As you rightly point out, there is no identified link between these two signatures which are correlative events. E2F is also correlated with a poor prognosis in NB due to a high number of aberrant cell divisions (Molenaar *et al.*, GCC), but to our knowledge there is no studies physically linking both events, just existing correlative studies (Strieder & Lutz, JBC). In order not to confuse the reader, we have moved the E2F signature to the additional data and we have modified the text in accordance with this comment (Expanded View 2A).

As you suggest and in order to definitely understand if the MYCN transcription factor is able to modify the expression of netrin-3, we carried out an analysis of the Netrin-3 level after having invalidated the expression of MYCN in two amplified MYCN neuroblastoma cell lines (IMR32 and IGRN91). We were thus able to observe a significant decrease of 70% in the transcript rate of *netrin-3* but not *netrin-1* in the absence of MYCN, which tends to confirm that MYCN directly regulates it (Figure 3C- Expanded View 2E).

8) The authors should consider including data on NTN1 from the CHIP-seq results presented in Figures 3 and 5 to show whether NTN3 and NTN1 share regulatory features

As suggested by the second reviewer, we performed a new analysis with Netrin-1 in the same CHIP-Seq experiments. As a result, we were unable to detect peaks in the netrin-1 gene sequence. This lack of binding probably explains at least in part the duality of expression in netrin-1 and netrin-3 in the SCLC and NB. These new results are now presented in the new supplementary Expanded View 3G and 4C.

9) The authors refer to Figure "2H" at the end of the section on NB - there is no such figure provided

We apologize for this mistake. The text has been corrected accordingly.

10) In the section on SCLC, the authors state that "netrin-3 expression is associated with the expression of two key transcription factors.." - this statement has not been proven and should be corrected

As noted by reviewer 2, we did not prove the direct links between these factors and netrin-3 in the previous version and we apologize for this lack of clarity. We first showed that both ASCL1 and NeuroD1 bind to the *netrin-3* promoter by CHIP sequencing analysis (Figure 5A and B). In order to determine whether the transcription factors ASCL1 and NeuroD1 could directly regulate the expression of *netrin-3* gene, we therefore conducted a new study in small cell lung cancer by silencing these two factors by siRNAs in NCI-H82 and NCI-H69 cells. We observed a decrease of the *netrin-3* transcript of 40% and 50% when ASCL1 and NeuroD1 were respectively silenced. These results are now presented in the new Figure 5 C.

11) In the section "Novel tumor growth promoting activity by netrin-3" the authors state that netrin-3 KO cells did not have differences in cell proliferation or death "(data not shown)" - this

data could easily be shown in a supplemental figure
These data has been now included in the new manuscript (Expanded view 5A).

12) Validation of the results in SCLC cells and tumors with NTN3 knockdown and NP137 in neuroblastoma models would dramatically enhance the impact of the manuscript

The number of cell lines for the *in vivo* experiments was increased. *In vivo* data now included 5 different models: IGRN91 and IMR32 for Neuroblastoma and NCI-H69, NCI-H82 and NCI-H2286 for small cell lung cancer.

To analyze the pro-tumoral role of netrin-3 *in vivo*, both IMR-32 and IGR-N91 NB cells were silenced with siRNAs for netrin-3 expression. Cells were xenografted on the chorioallantoic membrane (CAM) of ten-day-old chick embryos (Expanded View H). The CAM of chicken embryos is a well-described model to study NB tumor progression. Seventeen-day-old chick embryos were analyzed for primary tumor size. As shown in Figure 3 E-F-G, tumors derived from netrin-3 silenced cells were substantially smaller than tumors derived from mock-transfected cells in the two cell lines. These data clearly indicate that Netrin-3 is also a survival protein in neuroblastoma.

Additionally, we performed silencing both by siRNAs and CRISPr/Cas9 test on SCLC cells lines now presented in Expanded View 5A. The *in vivo* data on SCLC NCI-H82 and NCI-H2286 cell lines are now presented in Figure 7D - E and Expanded View 5E.

13) The authors only include images from single tumor samples in Figures 4B and Suppl Fig 1A - additional images from more samples (such as a tissue microarray) would be much more powerful, as Figure 4B in particular is not very convincing

As you probably know, SCLCs are one of the most proliferating solid cancers in humans. This particularity implies that they are very sensitive to chemotherapy and that the tumor mass almost completely disappears after treatment. Hence, they are never operated on with lung ablation. Samples are therefore very rare and there is no tissue microarray type collection, which is why we only had access to 13 samples (which are in fact ablations after misdiagnosis). New pictures have been provided with more cases (Expanded View 3B-C).

14) the impact of NP137 on SCLC xenograft tumors does not appear to be very dramatic - how do these effects compare to responses in NTN1-expressing tumors? The authors should consider including a comment on potential relative efficacy/response rates
We agree that the anti-cancer effects are not very impressive as a single agent but they are performed in very aggressive models. They are in line with the results obtained in several tumor types only expressing Netrin-1 and showing promising results (Grandin *et al*, Cancer Cell-2016). With these kind of results, antitumoral responses were observed in patients treated with anti-Netrin-1 during a Phase I study (Cassier *et al*; 2020). Our work will now focus on combining the antibody with standard of care in SCLC like Cisplatin to increase the efficacy of the compound. A comment has been added in the discussion to moderate the conclusion of the article.

15) how are the "Intake (%)" numbers calculated in Figure 6D and 6F?

We apologize for the lack of clarity in our explanations. Mice were implanted with either NCI-H82 or NCI-H69 SCLC cells; control or CRISPR-netrin-3 #1, #2, #3, by subcutaneous injection of 10^6 cells in 100 μ L of PBS into the right flank of mice. Tumor volume was calculated with the formula $V = (\text{length} \times \text{width}^2)/2$. A positive tumor catch is declared when the tumor reaches 100mm³. These analyses indicate that Netrin-3 probably plays a role very early at tumor onset. These experiments have to be compared with the experiments with tumors treated with the anti-netrin antibody which take place on already established tumors. The main text of the article and the materials and methods section have been modified accordingly.

16) the specific patient datasets used for Suppl Fig 2A, B are not clear and should be included in the figure (or figure legend).

This has been corrected in the new version of the manuscript. The patients used for this analysis in the present figure are the 181 patients used in Figure 3. They are dichotomized for NTN3 expression with/without MYCN oncogene for Suppl 2A. Figure 2B is the survival curve based on netrin-3 expression in patients of all stages but without MYCN amplification. The changes have been made and are now presented in the Expanded View 2C and D.

Referee #3 (Remarks for Author):

The manuscript by Jiang et al. investigates the potential role of netrin-3 in tumours. The authors find selected expression in two neuroectodermal tumour types, neuroblastoma (NB) and small cell lung cancer (SCLC). Expression of netrin-3 appears to be mutually exclusive with netrin-1 which has a broader expression pattern in cancer, and targeting of which is currently investigated in clinical trials.

Jiang et al. first establish the expression pattern of netrin-3 in neuroblastoma and its potential use as a prognostic marker in NB. Additionally, they briefly touch on potential mechanisms for transcriptional regulation of netrin-3 in NB and SCLC, mainly via ChIPSeq experiments. In the third part of the manuscript, the feasibility of targeting netrin-3 in SCLC is investigated. Using CRISPR/Cas9 deletion of netrin-3 or using an antibody against netrin-3 that cross-reacts with netrin-3 the authors demonstrate that netrin-3 plays a role in tumor-growth and/or initiation. In principle, the study provides some interesting data and targeting netrin-3 might provide an interesting therapeutic option. However, I feel the current manuscript touches upon several points without investigating each in sufficient depth to completely convince the reader. We thank reviewer 3 for his kind comments on our work, and we believe that we have now addressed all his/her points to be suitable for publication in EMBO Molecular Medicine.

Main points:

After the extensive analysis of netrin-3 in NB one would have expected that in vivo experiments targeting netrin-3 would also be performed with NB cell lines. This could add to the importance of netrin-3 expression in NB and the general applicability of netrin-3 targeting in cancer. In general, the number of cell lines used for the in vivo studies are quite low. A more extensive follow-up analysis of the in vivo experiments should be performed. Histology (H&E) with

stainings for e.g. Ki-67 could be informative. What is the suggested mechanism of action? For the experiments with netrin-3 deletion, the authors write that 'tumour development' is impaired. However, the experiments measure tumour take-rate which would be an indicator of clonogenicity. Why didn't the authors measure tumour growth in these experiments? Are there any molecular mechanisms the authors could propose? They state that neither proliferation nor apoptosis were changed in the netrin-3 deleted cells. It might be beneficial to show those important data, and also provide at least some idea what the underlying mechanism for the impaired *in vivo* growth is.

As highlighted by reviewer 3, the number of cell lines for the *in vivo* experiments was increased. *In vivo* data now include 5 different models: IGR-N91 and IMR-32 for Neuroblastoma and NCI-H69, NCI-H82 and NC-H2286 for small cell lung cancer.

To analyze the pro-tumoral role of Netrin-3 *in vivo*, both IMR-32 and IGR-N91 NB cells were silenced with siRNAs specific for netrin-3 expression and xenografted on the chorioallantoic membrane (CAM) of ten-day-old chick embryos (Figure 3E, F, G; Expanded View 2H-I-J). The CAM of chicken embryos is a well-described model to study NB tumor progression (Stupack *et al.* Nature). Seventeen-day-old chick embryos were analyzed for primary tumor size. As shown in Figures 3E-F, tumors derived from the netrin-3 silenced cells were substantially smaller than tumors derived from mock-transfected cells in the two cell lines. These data clearly indicate that Netrin-3 is also a survival protein in neuroblastoma.

To further explain the underlying mechanisms, we conducted Ki67, caspase-3 and PARP cleavage stainings of the engrafted tumors. As a consequence, an increase in cell death could be detected in tumors lacking Netrin-3 as compared to control tumors (Expanded View 2J). These results indicate that apoptosis is a central mechanism during Netrin-3 inhibition. In addition, we silenced Netrin-1 in the same cells and we did not detect significant modification of tumor growth. Additionally, we did not detect any modification of proliferation.

For the specific point of the CRISPR/Cas-9 clone tested in mice, it appears that some tumors deleted for Netrin-3 do not grow, or 20 days after the controls (Figure D and F), which is for us the most convincing result. It is therefore not possible to compare growth monitoring in the same timeframe when tumors do not grow in a group. It is probably for this reason that we have not identified any significant difference in the proliferation of CRISPR-Netrin-3 deleted clones at the experimental end-point, since once the tumor is implanted and in the growth phase the differences are erased in terms of cell division.

We therefore focused on tumors treated with the anti-Netrin antibody as these treatments are performed on already implanted tumors, as suggested by reviewer 3. We conducted an analysis of proliferation and cell death by testing the expression of the Ki67 division marker and activation of caspase-3 and the cleavage of PARP, two hallmarks of apoptosis (Figure 7F, Expanded View 5F). We performed this analysis by sampling the tumors two days after the 3rd treatment (*ie*: 400mm3). We were now able to detect an increase in cell apoptosis, determined by the PARP cleavage, in the anti-netrin treated group. On the other hand, we were unable to detect a decrease in proliferation under the conditions used. In terms of molecular mode of action, all these results indicate that Netrin-3 blockade impairs tumor progression by inducing

cell death rather than proliferation. These data have to be correlated with the literature and the inhibition

of netrin-1 by NP137 also induce cell death (Grandin *et al*, Cancer Cell-2016). Further studies will be need to understand the redundancy of these proteins in the impairing of apoptosis.

The analysis of transcription factor binding sites in the netrin-3 gene are somewhat correlative. A knockdown or knockout of the respective transcription factors would easily provide more meaningful data on the regulation of netrin-3 expression.

We totally agree with reviewer 3 and we believe we have now clarified this point. To determine the role of both MYCN, ASCL-1 and NeuroD-1 on the direct regulation of the *netrin-3* gene, we performed the depletion of these using specific siRNAs. As a consequence, we now show that netrin-3 expression decreases when MYCN is silenced in NB cell lines (chosen to be amplified for MYCN). This result appears to be specific to netrin-3 as we do not detect any modification in *netrin-1* gene expression. These data are presented in the new figure 3 C-D and Expanded view 4 A, B, C.

Moreover, we also observed a decrease in *netrin-3* gene expression when ASCL-1 and Neurod-1 were silenced in SCLC cell lines. These data and the Chip-seq analysis strongly argue in favor of a direct regulation of *netrin-3* by these transcription factors. The data are now presented in the new Figure 5A, B, C and Expanded View 4 A, B, C.

Additional points:

Figure 2E: The analysis of the validation cohort should also be performed separately for stage-4 patients (as in Fig. 4D).

A new analysis has been conducted only based on netrin-3 expression in aggressive Stage-4 patients. It confirms our previous observation that Netrin-3 expression predicts diagnosis in Stage 4 and is correlated with a poor survival (Expanded View 1C).

Figure 3: The GSEA appears to be performed on the whole cohort. As the authors demonstrate, netrin-3 expression correlates with stage and MYCN status the results are somewhat expected. An additional GSEA only in stage-4 netrin-3 high vs. low patients might provide more interesting results.

As suggested by reviewer 3, a new GSEA analysis has been carried out using the conditions requested by reviewer 3 and now presented in supplementary Expanded View 2 B. It confirms the data observed with the total cohort that Netrin-3 is highly expressed in poor prognosis NB with an enrichment in the hallmark G2/M checkpoint signature (Expanded View 2-B).

The legend to Figure 4 refers to a non-existing subfigure (q-RT-PCR)

We apologize for this mistake. The text has been removed.

Figure 6A: The figure is missing a legend for the colour keys in the affinity experiments.

Again, we apologize for this mistake. A color legend was added in the figure legend in the new version of the manuscript.

Fig. 6E: What is the netrin receptor expression on H69 cells?

The expression of netrin receptors in H69 cell line was added (Figure 6E).

The experiments for affinity measurements and the production of recombinant ligands are not described in the methods section.

This point has been added in the material and methods with a new bio layer interferometry section.

We are grateful to the reviewers for their comments, which we believe have strengthened the manuscript.

If we should send additional information, please let me know. We thank you in advance for your consideration of the revised manuscript.

15th Jan 2021

Dear Dr. Mehlen,

Thank you for the submission of your revised manuscript to EMBO Molecular Medicine, and please accept my apologies for the delay in getting back to you, which is due to the limited staff and increased submitted manuscripts during the holiday season. We have now received the enclosed reports from the 2 referees who re-reviewed your manuscript. As you will see, they are supportive of publication, and we will therefore be able to accept your manuscript pending the following final minor amendments:

1) Main manuscript text:

- Please answer/correct the changes suggested by our data editors in the main manuscript file (in track changes mode). This file will be sent to you in the next couple of days. Please use this file for any further modification.
- Please remove the green highlights.
- Please provide up to 5 keywords.
- Author Contribution: please complete (Valérie Combaret is missing). Peter Mulligan & Patrick Mehlen, and Maha Siouda & Misuaki Sanada need to be identified separately.
- Please remove "not shown" (p. 11). As per our guidelines on "Unpublished Data", the journal does not permit citation of "Data not shown". All data referred to in the paper should be displayed in the main or Expanded View figures.
- Material and Methods:
 - o Please include the "Supplementary Methods and Materials" in the Material and Methods section of your manuscript.
 - o Patients samples: Please include the full statement that informed consent was obtained from all subjects and the experiments conformed to the principles set out in the WMA Declaration of Helsinki and the Department of Health and Human Services Belmont Report.
 - o Cells: please indicate whether the cells were tested for mycoplasma contamination.
 - o RT-PCR: please provide the primers used for RT-PCR experiments
 - o Antibodies: please provide the dilutions for all antibodies.
- Data Availability Section: Before submitting your revision, primary datasets produced in this study need to be deposited in an appropriate public database (see <https://www.embopress.org/page/journal/17574684/authorguide#dataavailability>). The accession numbers and database should be listed in a formal "Data Availability" section (placed after Materials & Method). Please note that the Data Availability Section is restricted to new primary data that are part of this study. If no new primary dataset was produced, please indicate "This study includes no data deposited in external repositories"
- Funding: please complete funding information in the submission system as it has to match information provided in the manuscript. (MISSING: This work was supported by institutional grants from University of Lyon (PM), Centre Léon Bérard (PM), INSERM (PM) and CNRS (PM, BG). This work was also supported by grants from Fondation ARC for young investigators (BG), and from the Ligue Contre le Cancer (PM), SJ was supported by a LabEx DEVweCAN fellowship, MR was supported by la Ligue Contre le Cancer fellowship.)

2) Figures:

Please update all Figure EV callouts from eg: Expanded View 1 to Figure EV1

Figure 3D is not referenced in the main text, please update the callout.

Please add and define scale bars for Figure 4C, Figure EV1B, Figure EV2J, Figure EV3 B, C and Figure EV5F.

3) Source Data:

Thank you for providing Source Data. Please upload the Source Data files so as to have 1 pdf file per figure and 1 pdf file for EV figures. The first 2 pages of the Source Data are labeled Figure 6C but seem to match Figure EV2. Please clarify.

Source Data Figure 6C: Would you have blots where the ladder is included and clearly visible for UNC5B and netrin 1?

4) Checklist:

- Section E/12: Please include the full statement that informed consent was obtained from all subjects and the experiments conformed to the principles set out in the WMA Declaration of Helsinki and the Department of Health and Human Services Belmont Report.
- When a section does not apply to your study, please indicate N/A.
- Please fill in the section on Data Availability.

5) Please note that all corresponding authors are required to supply an ORCID ID for their name upon submission of a revised manuscript. While our submission system indicates that Patrick Mehlen is the corresponding author, the manuscript indicates both Patrick Mehlen and Benjamin Gibert, for whom there is no ORCID. Please clarify.

6) Please provide "The paper explained" section:

EMBO Molecular Medicine articles are accompanied by a summary of the articles to emphasize the major findings in the paper and their medical implications for the non-specialist reader. Please provide a draft summary of your article highlighting

7) Thank you for providing a nice synopsis picture. Please also provide a synopsis text that includes a short stand first (maximum of 300 characters, including space) as well as 2-5 one-sentences bullet points that summarizes the paper. Please write the bullet points to summarize the key NEW findings. They should be designed to be complementary to the abstract - i.e. not repeat the same text. We encourage inclusion of key acronyms and quantitative information (maximum of 30 words / bullet point). Please use the passive voice. Please attach these in a separate file or send them by email, we will incorporate them accordingly.

8) As part of the EMBO Publications transparent editorial process initiative (see our Editorial at <http://embomolmed.embopress.org/content/2/9/329>), EMBO Molecular Medicine will publish online a Review Process File (RPF) to accompany accepted manuscripts.

In the event of acceptance, this file will be published in conjunction with your paper and will include the anonymous referee reports, your point-by-point response and all pertinent correspondence relating to the manuscript. Let us know whether you agree with the publication of the RPF and as here, IF YOU WANT TO REMOVE OF NOT ANY FIGURES from it prior to publication.

I look forward to receiving your revised manuscript.

Yours sincerely,

Lise Roth

Lise Roth, PhD
Editor
EMBO Molecular Medicine

***** Reviewer's comments *****

Referee #2 (Remarks for Author):

The authors have addressed all of my prior comments and questions

Referee #3 (Remarks for Author):

The authors have adequately addressed my concerns and added a substantial amount of novel data that supports their conclusions. I now recommend this manuscript for publication.

The authors performed the requested editorial changes.

Thank you for sending the revised files. I have looked at everything and all is fine. I am thus very pleased to accept your manuscript for publication in EMBO Molecular Medicine!
It will be sent to our publisher to be included in the next available issue of EMBO Molecular Medicine.

Corresponding Author Name: Benjamin Gibert and Patrick Mehlen

Journal Submitted to: EMBO Mol Med

Manuscript Number: EMM-2020-12878